# Recognition through Reasoning: Reinforcing Image Geo-localization with Large Vision-Language Models

**Ling Li**[1]    **Yao Zhou**[3]    **Yuxuan Liang**[1]    **Fugee Tsung**[2,1]    **Jiaheng Wei**[1]*

[1]The Hong Kong University of Science and Technology (Guangzhou)
[2]The Hong Kong University of Science and Technology    [3]Independent Researcher
lli297@connect.hkust-gz.edu.cn, jiahengwei@hkust-gz.edu.cn

## Abstract

Previous methods for image geo-localization have typically treated the task as either classification or retrieval, often relying on black-box decisions that lack interpretability. The rise of large vision-language models (LVLMs) has enabled a rethinking of geo-localization as a reasoning-driven task grounded in visual cues. However, two major challenges persist. On the data side, existing reasoning-focused datasets are primarily based on street-view imagery, offering limited scene diversity and constrained viewpoints. On the modeling side, current approaches predominantly rely on supervised fine-tuning, which yields only marginal improvements in reasoning capabilities. To address these challenges, we propose a novel pipeline that constructs a reasoning-oriented geo-localization dataset, *MP16-Reason*, using diverse social media images. We introduce *GLOBE*, **G**roup-relative policy optimization for **L**ocalizability assessment and **O**ptimized visual-cue reasoning, yielding **B**i-objective geo-**E**nhancement for the VLM in recognition and reasoning. *GLOBE* incorporates task-specific rewards that jointly enhance localizability assessment, visual-cue reasoning, and geolocation accuracy. Both qualitative and quantitative results demonstrate that *GLOBE* outperforms state-of-the-art open-source LVLMs on geo-localization tasks, particularly in diverse visual scenes, while also generating more insightful and interpretable reasoning trajectories. The data and code are available at https://github.com/lingli1996/GLOBE.

## 1   Introduction

**The Background of Geo-localization.** The rapid growth of visual content on social media and mobile devices, has made image geo-localization (determining where an image was taken) increasingly important for downstream applications such as autonomous navigation [1, 2, 3] and crisis response [4]. Given that metadata (i.e., GPS coordinates) is frequently unavailable in practice [5], predicting geographic location from visual content remains a crucial capability. This demand has led to growing interest in the image geo-localization task [6].

**Limitations in Existing Geo-localization Approaches.** Traditional image geo-localization approaches fall into two main categories: classification and retrieval. Classification-based methods [7, 8, 9, 10, 11] treat geo-localization as a discrete prediction task, assigning each image to a predefined set of geographical regions or cells. Retrieval-based methods [12, 13, 14, 15, 16, 17, 18] estimate location by comparing the query image to a large geo-tagged reference database, retrieving the closest match in terms of visual features, geographic coordinates, or semantic labels (e.g., city or country names). In practice, retrieval-based approaches can achieve higher accuracy at fine-grained precision [19, 20], making them particularly effective when exact localization is required. Although

---

*Corresponding Author: jiahengwei@hkust-gz.edu.cn

these methods perform well on standard benchmarks, they typically require training on millions of samples and lack interpretability, offering little insight into their underlying reasoning process.

**When LVLMs Meet Geo-localization.** The emergence of Large Vision-Language Models (LVLMs) [21, 22, 23, 24, 25, 26, 27] has introduced a new paradigm to tackle image geo-localization. Equipped with powerful multimodal reasoning capabilities and extensive world knowledge encoded through large-scale pretraining, LVLM-based methods [28, 19, 29] have been explored through various strategies, including few-shot prompting, retrieval-augmented generation (RAG), and supervised fine-tuning (SFT). These methods are capable of generating both location predictions and explanations, offering greater interpretability in how decisions are made.

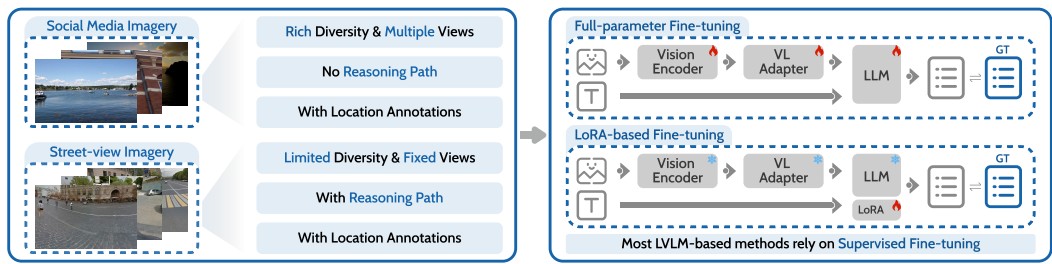

Figure 1: Overview of data and modeling limitations in LVLM-based image geo-localization.

**Limitations in LVLM-based Image Geo-localization.** Notably, geo-localization requires **deeper reasoning** than typical vision-language tasks. Success depends on **more than recognition**, as models must often draw on domain knowledge to infer plausible locations from subtle visual cues such as vegetation, architecture, or language, especially when iconic landmarks are absent. While LVLMs offer a promising path toward such reasoning-driven geo-localization, two fundamental challenges persist, as illustrated in Figure 1. **On the data side**, existing datasets rarely provide explicit reasoning supervision, such as interpretations of visual evidence and contextual justifications supporting the final location decision. Recent efforts [28, 30, 31] to incorporate reasoning into geo-localization datasets have primarily relied on street-view imagery, which offers limited scene diversity and fixed viewpoints. As a result, models trained on such data often struggle to generalize to diverse, real-world visual conditions. **On the modeling side**, most current approaches depend on supervised fine-tuning with instruction-style data, which tends to encourage pattern replication rather than the development of a grounded understanding of visual-geographic relationships. Without verification mechanisms, these models rely heavily on correlation rather than structured inference, reducing their ability to generalize beyond familiar examples.

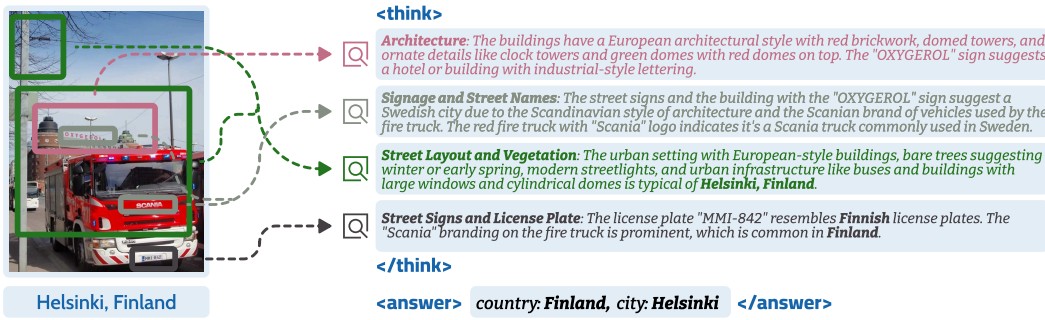

Figure 2: Example reasoning trajectories generated by *GLOBE*, illustrating interpretable and visually grounded geolocation predictions.

**How *GLOBE* Tackles the Challenges.** To address these challenges, we propose a novel pipeline for reasoning-driven geo-localization consisting of two main components: (1) constructing a geo-localization dataset from diverse social media images augmented with model-derived reasoning traces, and (2) fine-tuning a vision-language model using Group Relative Policy Optimization (GRPO) for enhanced reasoning. We begin by building *MP16-Reason*, an extension of MP-16 [32], which contains user-captured photographs with diverse viewpoints and rich contextual content. To

introduce reasoning supervision, we prompt multiple vision-language models [24, 33, 15] to distill the geolocation-related knowledge, including localizability assessments, reasoning trajectories, and predicted locations. To ensure the reliability of these distilled signals, we employ a multi-dimensional verification process that assesses both the alignment between visual evidence and model-generated reasoning, and the consistency across different models through self-verification, thereby filtering out inconsistent or hallucinated outputs. Finally, we fine-tune a pretrained LVLM on the curated dataset using GRPO [34], guided by task-specific rewards for localizability, visual grounding, and geolocation accuracy. Our resulting model, *GLOBE*, achieves state-of-the-art performance among open-source VLMs on geo-localization benchmarks, while producing more interpretable and visually grounded reasoning trajectories, as shown in Figure 2. Our main contributions include:

- **Reasoning-Oriented Geo-Localization Dataset:** We construct *MP16-Reason*, a diverse geo-localization dataset enriched with image-grounded reasoning supervision that supports model interpretability and generalization.

- **GRPO-Based Fine-Tuning:** We develop a GRPO-based reinforcement learning framework that fine-tunes LVLMs using task-specific rewards for localizability, visual grounding, and geolocation accuracy, enabling stronger reasoning capabilities compared to traditional supervised fine-tuning.

- **Opensource LVLM:** Trained through this pipeline, we opensource *GLOBE*. Empirical results demonstrate that *GLOBE* outperforms state-of-the-art LVLMs on multiple geo-localization benchmarks, while producing more interpretable and visually grounded reasoning trajectories.

## 2  Related Work

**Image Geo-localization.** Image geo-localization aims to predict the geographic location of a given image and has broad applications in urban analysis [35, 4, 36, 37, 38], navigation [1], and geospatial data mining [39, 40, 41, 42, 43]. General methods like Visual Place Recognition (VPR) [44, 45, 46, 47] focus on robustness to challenging variations (e.g., illumination and viewpoint). With advances in multimodal models, research has evolved from classification [7, 8, 9, 10, 11] and retrieval-based methods [12, 13, 14, 15, 16, 17, 18] to generation-based approaches [48, 28, 29, 19, 49], which aim to produce location predictions through visual reasoning. Recent studies [28, 29, 19] have pointed out key limitations of classification (e.g., coarse granularity) and retrieval methods (e.g., dependency on large reference databases), prompting increased interest in generation-based alternatives. Since the introduction of the MediaEval Placing Tasks 2016 (MP-16) dataset by [32], recent research [29, 19] continues to utilize this dataset to model relationships between visual semantics and geographic locations. In contrast to conventional approaches, current LVLMs [21, 22, 23, 24, 25], which are typically pre-trained on large-scale datasets, inherently exhibit significant visual reasoning capabilities. This raises the critical question of **whether the continued reliance on millions of labeled samples for supervised fine-tuning remains necessary to effectively adapt these models to specific tasks**. In this work, we take a data-centric perspective to explore how large-scale datasets can be used to build higher-quality training data for fine-tuning LVLMs in image geo-localization.

**Large Vision-Language Models.** Building on recent LLM advancements [50, 51, 52, 53, 54, 55, 56], LLaVA [21] demonstrated that combining a vision encoder with an LLM and jointly fine-tuning them improves image-based question answering [57, 58, 59, 60]. Subsequently, various LVLMs have emerged [22, 23, 24, 25, 26, 27], differing primarily in their visual-language alignment mechanisms and associated architectural trade-offs. Motivated by these recent advancements, our work further investigates the shift of image geo-localization from traditional methods to LVLMs. **Specifically, we explore how curated datasets can be effectively leveraged to facilitate more efficient fine-tuning of these models for geo-localization tasks.**

**Visual Reasoning and Verification.** The emergence of advanced models such as DeepSeek [61] has heightened expectations for the multimodal reasoning capabilities of LLMs. Most reasoning research [62, 63] has focused on mathematical tasks, with limited attention to open-ended or visual scenarios. Thus, these models often suffer from hallucination [64, 65, 66], especially in visual tasks where they produce seemingly plausible but incorrect outputs. To address hallucination and promote more faithful reasoning, recent work has explored verification-based strategies [67, 68, 69, 70, 71], as well as reinforcement learning frameworks [34, 72] that optimize models via structured rewards. **Motivated by these insights, we adopt GRPO as the reinforcement learning framework in our reasoning-driven geo-localization task.**

# 3  🌐 *GLOBE*: The Methodology

We propose a novel pipeline based on the original MP-16 [32] dataset, aiming to advance image geo-localization from single-modal visual recognition to more robust multimodal reasoning. Achieving this objective requires not only powerful models but also well-curated training data that effectively capture geographic cues. Our pipeline for reasoning-driven geo-localization consists of two main components: dataset curation and model fine-tuning. These are implemented in three stages: (1) dataset curation via strong-to-weak distillation & verification (Section 3.1), (2) reward construction via task-specific supervision (Section 3.2), and (3) model fine-tuning via GRPO-based reinforcement learning (Section 3.3).

## 3.1  Dataset Curation: Data Distillation & Verfication

Raw web-scale datasets contain a diverse range of social media images captured from varied perspectives. However, these datasets suffer from substantial noise [73, 74, 75, 76, 77], such as close-up shots with limited visual context or generic objects lacking informative localizable cues. To address this issue and select appropriate images for downstream training, we employ multi-model vision-language knowledge distillation for data synthesis and multi-dimensional verification for data curation.

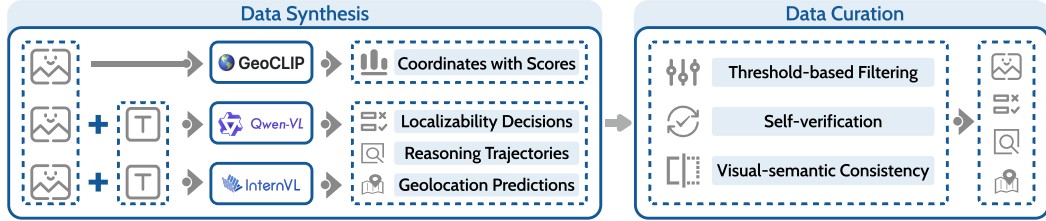

Figure 3: The pipeline of data synthesis and curation via multi-model distillation and verification.

**Multiple Vision-Language Models Knowledge Distillation.** We utilize multiple vision-language models (e.g., Qwen2.5-VL-72B [24], InternVL3-78B [33], and GeoCLIP [15]) to extract localizability judgments, visual cues, and geolocation predictions for each image in the MP-16 [32] dataset, inspired by [48, 28, 78]. *The use of three diverse, high-performing VLMs is a deliberate design choice to mitigate model-specific biases.* Rather than relying on a single model, which may reflect its own systematic preferences or reasoning patterns, combining multiple VLMs allows us to leverage their consensus and complementarity, thereby enhancing both the robustness and diversity of the distilled signals [79]. As shown in Figure 3, Qwen2.5-VL-72B [24] and InternVL3-78B [33] produce binary localizability decisions, step-by-step reasoning trajectories, and textual geolocation predictions. GeoCLIP [15], in contrast, produces latitude-longitude coordinates along with a confidence score that quantifies localizability [78]. Collectively, these strong models offer complementary signals, which we distill into structured supervision for downstream data curation and reward modeling.

**Multi-dimensional Verification.** Following model inference, we perform multi-dimensional verification to curate high-quality data, as illustrated in Figure 3. Initially, we filter out images with negative localizability decisions or low localizability scores. Subsequently, incorrect geolocation predictions are discarded by comparing them against ground-truth annotations. To ensure the reliability of the knowledge distilled from Qwen2.5-VL-72B [24] and InternVL3-78B [33], we introduce a self-verification step in which the geolocation predictions and reasoning trajectories of both models are compared for each image. Only those samples exhibiting consistent location outputs (e.g., matching city- or country-level predictions) and semantically aligned reasoning chains are retained. This cross-model agreement serves as the reliability proxy in distilled supervision. Furthermore, to enforce visual grounding of the reasoning process, we employ a general-purpose semantic segmentation model [80] to extract both the categories and relative proportions of visual elements within each image. The segmentation produces pixel-level labels across a wide range of semantic categories (e.g., sky, building, road, vegetation, and car), providing a dense understanding of the scene composition. We then assess the consistency between the entities mentioned in the reasoning trajectories and the detected visual elements, ensuring that the reasoning is supported by actual visual evidence rather than coincidental correlations. Through this multi-stage validation pipeline, which combines localizability filtering, self-verification of distilled knowledge, and visual-semantic consistency checks, we curate a robust and trustworthy dataset tailored for downstream tasks.

## 3.2 Reward Construction: Task-specific Supervision

Building upon the curated dataset introduced in Section 3.1, we develop three task-specific rewards to assess distinct dimensions of reasoning quality in the geo-localization process. Each reward is trained with annotated supervision and collectively provides a structured reward signal, which guides the policy optimization during the reinforcement learning stage described in Section 3.3.

Formally, let $\mathcal{D} = (I_i, y_i, g_i, r_i)_{i=1}^N$ denote the curated dataset of $N$ samples, where $I_i$ is an image, $y_i \in \{0, 1\}$ is a binary label indicating whether the image is localizable, $g_i$ indicates the ground-truth geolocation, and $r_i$ is the associated reasoning trajectory.

**Localizability Reward.** We introduce a localizability reward to estimate how well an image, together with its predicted reasoning $\hat{r}_i$, can support reliable localization. In other words, localizability reflects the joint contribution of the visual content and the reasoning process to the likelihood of correct localization. To this end, we train a LLM-based reward model on the curated dataset $\mathcal{D}$, where the objective is to distinguish whether a given pair $(I_i, \hat{r}i)$ corresponds to a localizable case ($y_i = 1$). Instead of using only the image as input, incorporating the predicted reasoning allows the model to exploit semantic cues that indicate the interpretability and consistency of the localization process. Formally, the reward is defined as:

$$R_{\text{loc}}(I_i, \hat{r}_i) = \mathbb{P}(y_i = 1 \mid I_i, \hat{r}_i; \theta_{\text{loc}}), \tag{1}$$

where $\theta_{\text{loc}}$ denotes the parameters of the reward model. The resulting probability score serves as a reward signal for reinforcement learning and as a soft indicator of the localizability of the image–reasoning pair.

**Visual Grounding Consistency Reward.** To ensure the model-generated reasoning aligns with the actual visual content, we introduce a reward model that evaluates entity grounding consistency. For a given sample $(I_i, r_i)$ from the curated dataset, let $\hat{r}_i$ denote the predicted reasoning. We extract a set of entities $E_i = \{e_1, e_2, ..., e_n\}$ from the reasoning trajectory $\hat{r}_i$, and a set of visual elements $V_i = \{v_1, v_2, ..., v_m\}$ from both the image $I_i$ (via semantic segmentation) and the text of $r_i$ (via entity extraction). We define a soft matching function $\text{Match}(e_j, V_i) \in 0, 1$, which returns 1 if entity $e_j$ approximately matches any element in $V_i$, allowing for partial lexical or semantic overlap. The visual grounding reward is computed as:

$$R_{\text{vis}}(I_i, \hat{r}_i, r_i) = \frac{1}{|E_i|} \sum_{j=1}^{|E_i|} \text{Match}(e_j, V_i), \tag{2}$$

where $R_{\text{vis}}$ assigns a higher score when more entities in the reasoning are visually grounded. This reward penalizes hallucinated entities that do not correspond to visible elements in the image, thereby encouraging grounded visual reasoning.

**Geo-localization Accuracy Reward.** To evaluate model predictions at a semantic location level, we define a classification-based reward that reflects whether the predicted country and city match the ground truth. Let $\hat{g}_i = (\hat{c}_i, \hat{t}_i)$ denote the predicted country and city for image $I_i$, and let $g_i = (c_i, t_i)$ be the corresponding ground-truth geolocation from the curated dataset. The geo-localization reward $R_{\text{geo}}$ is defined as:

$$R_{\text{geo}}(\hat{g}_i, g_i) = \mathbb{I}[\hat{c}_i = c_i] \cdot \left( \alpha \cdot \mathbb{I}[\hat{t}_i = t_i] + (1 - \alpha) \right), \tag{3}$$

where $\mathbb{I}[\cdot]$ is the indicator function and $\alpha \in [0, 1]$ is a weighting factor that controls the importance of city-level correctness, conditional on the country being correct. This reward structure captures the hierarchical nature of geo-tags. A reward of 0 is assigned when the predicted country is incorrect (i.e., $\hat{c}_i \neq c_i$). If the country is correct but the city is not (i.e., $\hat{c}_i = c_i, \hat{t}_i \neq t_i$), the model receives a partial reward of $1 - \alpha$. A full reward of 1 is assigned only when both predictions are correct (i.e., $\hat{c}_i = c_i, \hat{t}_i = t_i$). This tiered design encourages the model to first learn coarse-grained localization before refining its predictions to finer spatial resolutions.

## 3.3 Model Fine-tuning: GRPO-based Reinforcement Learning

With the reward signals defined in Section 3.2, we fine-tune the base model using GRPO [34], a reinforcement learning algorithm designed for ranking-based reward optimization, as illustrated

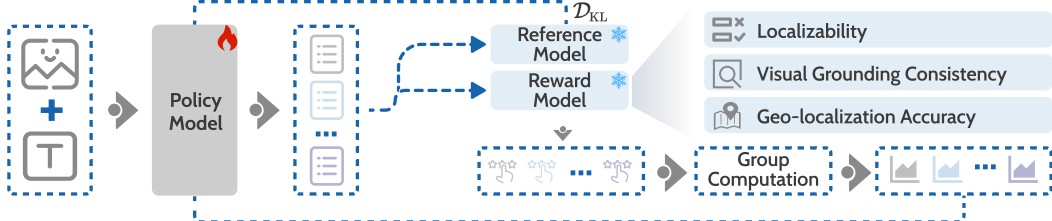

Figure 4: GRPO optimization framework with multi-dimensional reward design. For each prompt, candidate outputs are scored using three task-specific reward models: $R_{\text{loc}}$, $R_{\text{vis}}$, and $R_{\text{geo}}$, which reflect different aspects of geo-localization reasoning. Group-wise advantage values guide policy updates, while a $\mathcal{D}_{\text{KL}}$ penalty constrains divergence from the reference model.

in Figure 4. GRPO builds upon Proximal Policy Optimization (PPO)[81], which stabilizes policy updates by optimizing a clipped surrogate objective using advantage estimates derived from scalar rewards. Unlike PPO, GRPO introduces group-wise normalization and optimizes relative preferences among candidates conditioned on each prompt, enhancing robustness to variations in the reward scale.

Let $\pi_\theta$ denote the current policy parameterized by $\theta$, and let $\mathcal{B} = \{(\boldsymbol{x}_i, \{\boldsymbol{a}_i^{(j)}\}_{j=1}^k)\}$ represent a batch of input prompts $\boldsymbol{x}_i$ each paired with $k$ candidate completions $\boldsymbol{a}_i^{(j)}$ sampled from the policy. Each completion $\boldsymbol{a}_i^{(j)}$ is scored by a composite reward function:

$$r_i^{(j)} = \lambda_1 R_{\text{loc}} + \lambda_2 R_{\text{vis}} + \lambda_3 R_{\text{geo}}, \tag{4}$$

where $\lambda_1, \lambda_2, \lambda_3 \in [0, 1]$ are weights controlling the importance of the three reward components: localizability ($R_{\text{loc}}$), visual grounding consistency ($R_{\text{vis}}$), and geo-localization accuracy ($R_{\text{geo}}$).

To encourage the model to prefer higher-reward completions within each group, GRPO computes a group-normalized advantage for each candidate:

$$A_i^{(j)} = \frac{r_i^{(j)} - \mu_i}{\sigma_i}, \quad \mu_i = \frac{1}{k} \sum_{l=1}^k r_i^{(l)}, \quad \sigma_i = \sqrt{\frac{1}{k} \sum_{l=1}^k \left(r_i^{(l)} - \mu_i\right)^2}, \tag{5}$$

which centers rewards within each prompt group. Eqn. (5) guides the policy to optimize relative ranking rather than absolute scores, making it suitable for scenarios with non-uniform reward scales.

The policy is then updated by maximizing the following clipped surrogate objective:

$$\mathcal{L}_{\text{GRPO}}(\theta) = \mathbb{E}_{(\boldsymbol{x}_i, \boldsymbol{a}_i^{(j)}) \sim \pi_{\theta_{\text{ref}}}} \left[ \min\left(\rho_i^{(j)} A_i^{(j)}, \text{clip}(\rho_i^{(j)}, 1 - \epsilon, 1 + \epsilon) A_i^{(j)}\right) - \beta \mathcal{D}_{\text{KL}}\left[\pi_\theta \| \pi_{\text{ref}}\right] \right], \tag{6}$$

where $\rho_i^{(j)} = \frac{\pi_\theta(\boldsymbol{a}_i^{(j)}|\boldsymbol{x}_i)}{\pi_{\theta_{\text{old}}}(\boldsymbol{a}_i^{(j)}|\boldsymbol{x}_i)}$ is the likelihood ratio between the current and reference policies, and $\epsilon$ is the clipping threshold. The coefficient $\beta$ controls the strength of the $\mathcal{D}_{\text{KL}}$ penalty, and $\pi_{\text{ref}}$ is the reference policy used to constrain updates. In practice, the reference policy $\pi_{\text{ref}}$ is typically instantiated as the previous policy snapshot, serving to regularize updates and ensure training stability.

# 4   Experiments

We conduct both qualitative and quantitative experiments, including ablation studies, to evaluate the effectiveness of our curated dataset *MP16-Reason* and the GRPO-based training strategy employed in *GLOBE*. Specifically, we examine whether *MP16-Reason* enables better geo-reasoning (i.e., the ability to infer geographic locations through interpretable and visually grounded reasoning) compared to conventional image-only datasets (which lack reasoning supervision) and street-view datasets (which offer limited visual diversity). We further assess whether GRPO training provides stronger reasoning capability than supervised fine-tuning, and compare *GLOBE* against both open- and closed-source LVLMs.

## 4.1 Experimental Setup

**Datasets.** The curated dataset *MP16-Reason* is divided into two subsets: *MP16-Reason*-Train with 33k samples and *MP16-Reason*-Test with 12k samples, respectively. *MP16-Reason*-Train is used to train *GLOBE*, while *MP16-Reason*-Test is used to evaluate all baseline methods. The detailed statistics of these subsets, including sample size, geographic coverage, and scene distribution, are summarized in Table 1. Notably, *MP16-Reason*-Test was deliberately constructed to cover a broader geographic range (e.g., more countries and cities), including locations not present in the training set, which allows evaluation of how well the model generalizes in geo-reasoning beyond the training distribution. To ensure a comprehensive comparison, we additionally evaluate all models on the public geo-localization benchmark IM2GPS3K [82] and OSV-5M [83].

Table 1: Statistics of the proposed *MP16-Reason*.

| Dataset | #Samples | #Country | #City | #Indoor Scene | #Natural Scene | #Urban Scene |
|---|---|---|---|---|---|---|
| *MP16-Reason*-Train | 33721 | 134 | 1944 | 5393 | 2077 | 26251 |
| *MP16-Reason*-Test | 12000 | 145 | 3012 | 2096 | 1092 | 8812 |

# denotes the number of instances.

**Evaluation Metrics.** We follow previous work [15, 16, 19, 28] and report the percentage of predictions whose geographic distance to the ground-truth coordinate falls within fixed thresholds (1km, 25km, 200km, 750km, and 2500km). Since our model outputs discrete place names (e.g., country or city), we concatenate the predicted city and country into a single string and query Microsoft Azure Maps [1], which returns the corresponding representative GPS coordinate (e.g., the geographic center of the region) for evaluation.

**Implementation details.** For data curation, we deployed Qwen2.5-VL-72B and InternVL3-78B using $8 \times$ H20 GPUs under the VLLM framework, while GeoCLIP was run separately on a single H20 GPU. These models were used to perform inference over the original MP16 dataset. We then built *GLOBE* on top of Qwen2.5-VL-7B [24], a publicly available LVLM with strong multimodal understanding capabilities. Instead of using task-specific supervised fine-tuning as a cold start, we directly fine-tune the model using reinforcement learning based on the GRPO framework described in Section 3.3. In GRPO training, the 7B model was trained on $8 \times$ H20 GPUs with a batch size of 16, yielding a throughput of approximately 0.44 examples per second. Further implementation details are provided in Appendix A.1.

## 4.2 Experimental Results

To assess the impact of the curated dataset and the proposed training strategy, we conduct both external baseline comparisons and internal ablation studies, as detailed in the following subsections.

### 4.2.1 Baseline Comparison

We evaluate the geo-localization performance of *GLOBE* on both *MP16-Reason*-Test and the public benchmark IM2GPS3K [82] (see Table 2). For clarity, we organize the baselines into three categories: (I) *Image-only supervision*, which relies purely on visual features and coordinate labels without reasoning signals; (II) *Open- and closed-source LVLMs*, including general-purpose LVLMs trained on diverse multimodal data; and (III) *Task-specific reasoning supervision*, which refers to models trained on geo-localization datasets with reasoning-oriented annotations, often dominated by street-view imagery. We further assess generalization on the street-view dataset OSV-5M [83] (mini-3K), with detailed results provided in Appendix A.2.2. In addition, we examine performance under different scene conditions to demonstrate the robustness of *GLOBE*, as reported in Appendix A.2.1.

**Image-only supervision.** Compared with models trained solely on large-scale image-only supervision (e.g., MP-16 with over 4M samples), *GLOBE* achieves comparable or even superior accuracy using only 33K samples from *MP16-Reason*. This efficiency gain stems from reasoning-driven supervision, which provides explicit localizability judgments and visual grounding signals beyond raw coordinates. These findings suggest that reasoning annotations can substantially compensate for data scale, enabling more data-efficient geo-localization.

---

[1]https://portal.azure.com/

Table 2: Geo-localization performance comparison on *MP16-Reason*-Test and IM2GPS3K [82].

| Method | Dataset, Size | *MP16-Reason*-Test (% @ km) | | | | | IM2GPS3K [82] (% @ km) | | | | |
|---|---|---|---|---|---|---|---|---|---|---|---|
| | | Street 1km | City 25km | Region 200km | Country 750km | Continent 2500km | Street 1km | City 25km | Region 200km | Country 750km | Continent 2500km |
| **I. Image-only supervision** | | | | | | | | | | | |
| ISNs [9] | MP-16, 4M | 26.24 | 47.38 | 55.88 | 68.48 | 80.92 | 10.50 | 28.00 | 36.60 | 49.70 | 66.00 |
| GeoCLIP [15] | MP-16, 4M | 29.28 | 52.52 | 66.85 | 84.07 | 93.33 | 14.11 | 34.47 | 50.65 | 69.67 | 83.82 |
| Translocator[†] [10] | MP-16, 4M | - | - | - | - | - | 11.80 | 31.10 | 46.70 | 58.90 | 80.10 |
| PIGEOTTO[†] [16] | MP-16, 4M | - | - | - | - | - | 11.30 | 36.70 | 53.80 | 72.40 | **85.30** |
| G3 (GPT4V)[†] [19] | MP-16, 4M | - | - | - | - | - | **16.65** | 40.94 | 55.56 | 71.24 | 84.68 |
| Hybrid [83] | OSV-5M, 5M | 0.97 | 16.53 | 28.72 | 50.31 | 71.47 | 0.83 | 13.28 | 25.33 | 43.84 | 65.63 |
| RFM-YFCC [49] | Flickr, 48M | 11.72 | 46.64 | 60.46 | 77.97 | 91.96 | 5.41 | 29.70 | 44.71 | 61.83 | 79.55 |
| **II. Open- and closed-source LVLMs** | | | | | | | | | | | |
| Qwen2.5-VL-7B [24] | - | 15.42 | 52.72 | 62.86 | 75.11 | 83.47 | 8.58 | 32.53 | 43.11 | 58.93 | 72.37 |
| InternVL3-8B [33] | - | 12.01 | 44.17 | 55.66 | 75.36 | 86.98 | 6.44 | 25.69 | 34.57 | 49.38 | 61.66 |
| Gemma3-27B [84] | - | 16.03 | 55.63 | 68.07 | 82.59 | 91.29 | 8.48 | 33.37 | 46.61 | 63.63 | 79.95 |
| InternVL3-78B [33] | - | 14.72 | 52.46 | 65.25 | 81.73 | 91.17 | 8.93 | 35.05 | 47.32 | 64.03 | 78.64 |
| Qwen2.5-VL-72B [24] | - | 17.52 | 59.30 | 71.01 | 84.06 | 91.65 | 9.11 | 35.77 | 48.35 | 64.96 | 78.88 |
| Doubao1.5-VL[†] [85] | - | 18.89 | 64.02 | 76.55 | 88.33 | 93.44 | 11.61 | 46.21 | **60.60** | **75.04** | 85.09 |
| GPT-4.1[†] [86] | - | **20.05** | **66.76** | **79.70** | **89.84** | **94.53** | 12.11 | **46.85** | 60.36 | 74.41 | 85.25 |
| **III. Task-specific reasoning supervision** | | | | | | | | | | | |
| GeoReasoner-7B [28] | GSV, 133K | 10.06 | 40.44 | 50.91 | 68.01 | 79.68 | 7.67 | 26.94 | 36.63 | 52.27 | 65.39 |
| GaGA[†] [30] | MG-Geo, 5M | - | - | - | - | - | 11.70 | 33.00 | 48.00 | 67.10 | 82.10 |
| *GLOBE*-7B (Ours) | MP16-Reason, 33K | 17.99 | 62.85 | 73.83 | 86.68 | 92.52 | 9.84 | 40.18 | 56.19 | 71.45 | 82.38 |

† denotes models that are not publicly available. Underlined results indicate test–train overlap. Best open- and closed-source results are in blue and **bold**, respectively.

**Open- and closed-source LVLMs.** *GLOBE* achieves stronger results than open-source LVLMs, outperforming much larger models such as Qwen2.5-VL-72B [24] and InternVL3-78B [33]. Notably, GLOBE, built on the Qwen2.5-VL-7B [24] backbone, surpasses Qwen2.5-VL-72B [24], the larger model originally used to generate the distilled annotations. This outcome highlights the effectiveness of our distillation and GRPO-based training framework in extracting and refining knowledge rather than merely replicating model outputs. In addition, qualitative comparisons of reasoning trajectories are provided in Appendix A.2.5, further illustrating the interpretability advantages of *GLOBE*. Compared with closed-source industrial systems such as Doubao1.5-VL [85] and GPT-4.1 [86], *GLOBE* remains behind. This gap is expected, as the training data scale and settings of these systems are not publicly disclosed. We aim to advance open, reproducible, and data-efficient LVLM training to support sustainable progress.

**Task-specific reasoning supervision.** Relative to models trained on task-specific reasoning datasets dominated by street-view imagery, *GLOBE* demonstrates stronger robustness. By incorporating different types of scenes during the construction of *MP16-Reason*, our approach achieves superior generalization, particularly when evaluated under diverse scene conditions (see Appendix A.2.1). Under comparable 7B backbones, *GLOBE* consistently outperforms counterparts (Qwen2.5-VL-7B [24] and GeoReasoner-7B [28]), confirming the necessity of scene diversity for real-world geo-localization.

We further evaluate on OSV-5M [83] (mini-3K), a street-view dataset outside the training domain of *MP16-Reason* (see Appendix A.2.2). Despite this domain shift, *GLOBE* surpasses open-source methods such as ISNs [9] and GeoCLIP [15], which are trained on data distributions similar to *MP16-Reason*, as well as counterparts such as Qwen2.5-VL-7B [24] and InternVL3-8B [33]. These results demonstrate that reasoning-driven supervision enhances in-domain performance while enabling superior generalization to unseen domains. Representative failure cases are discussed in Appendix A.2.3, providing qualitative insights into the model's limitations. Beyond accuracy and generalization, we also provide an efficiency comparison in Appendix A.2.4.

### 4.2.2 Ablation Study

To better understand the contributions of our design choices, we conduct ablation studies along three dimensions: (I) the *reward components* used in GRPO training; (II) the *backbone models* on which our method is applied; and (III) the *distillation datasets* employed for supervision. These experiments allow us to disentangle the effects of supervision signals, model capacity, and data quality, thereby providing a more comprehensive view of the strengths of *GLOBE* and *MP16-Reason*.

**Reward components.** Table 3 presents ablation results for GRPO training under different reward configurations, including Localizability (Loc), Visual Grounding Consistency (VGC), and Geo-localization Accuracy (GA). Using all three rewards yields the highest overall performance (row 9). Removing any single component (rows 6-8) causes noticeable drops, highlighting the importance of

Table 3: Ablation on reward components with Qwen2.5-VL-7B [24] backbone.

| Model | CoT | SFT | Loc Reward | GRPO VGC Reward | GA Reward | Street 1km | City 25km | Region 200km | Country 750km | Continent 2500km |
|---|---|---|---|---|---|---|---|---|---|---|
| | | | | | | | | *MP16-Reason*-Test (% @ km) | | |
| Qwen2.5-VL-7B [24] | | | | | | 14.37 | 51.11 | 61.29 | 73.67 | 82.46 |
| Qwen2.5-VL-7B [24] | ✓ | | | | | 15.42 | 52.72 | 62.86 | 75.11 | 83.47 |
| Qwen2.5-VL-7B [24] | ✓ | ✓ | | | | 16.38 | 56.76 | 70.21 | 83.82 | 90.75 |
| *GLOBE* w/o Loc&GA | ✓ | | | ✓ | | 17.01 | 59.36 | 71.77 | 84.44 | 91.76 |
| *GLOBE* w/o Loc&VGC | ✓ | | | | ✓ | 17.24 | 59.24 | 71.93 | 84.69 | 91.54 |
| *GLOBE* w/o Loc | ✓ | | | ✓ | ✓ | 17.50 | 59.58 | 71.23 | 84.06 | 91.23 |
| *GLOBE* w/o VGC | ✓ | | ✓ | | ✓ | 17.52 | 59.83 | 72.22 | 84.72 | 91.12 |
| *GLOBE* w/o GA | ✓ | | ✓ | ✓ | | 17.44 | 59.53 | 71.41 | 84.33 | 91.18 |
| *GLOBE* | ✓ | | ✓ | ✓ | ✓ | 17.99 | 62.85 | 73.83 | 86.68 | 92.52 |

Best results are in blue.

reasoning-driven supervision beyond coordinate accuracy alone. Moreover, GRPO outperforms SFT (row 9 vs. row 3), delivering stronger consistency and grounding by leveraging reward signals to guide output quality. Even with partial reward combinations, GRPO still surpasses SFT, demonstrating the clear advantage of reinforcement learning with reasoning-driven supervision. In addition to the choice of reward components, the weighting hyperparameters $\lambda_1$, $\lambda_2$, and $\lambda_3$ in the GRPO objective also play a role in balancing supervision. A detailed discussion of their design rationale and experimental evaluation is provided in Appendix A.2.6, while an analysis of reward trajectories throughout the training process is presented in Appendix A.2.7.

Table 4: Ablation on backbone architectures.

| Backbone | Training Strategy | Street 1km | City 25km | Region 200km | Country 750km | Continent 2500km |
|---|---|---|---|---|---|---|
| | | | | *MP16-Reason*-Test (% @ km) | | |
| InternVL3-8B [33] | Baseline | 12.01 | 44.17 | 55.66 | 75.36 | 86.98 |
| | SFT | 12.41 | 44.68 | 56.37 | 75.20 | 86.32 |
| | GRPO | 17.47 | 60.09 | 72.41 | 85.02 | 91.92 |
| Qwen2.5-VL-7B [24] | Baseline | 15.42 | 52.72 | 62.86 | 75.11 | 83.47 |
| | SFT | 16.38 | 56.76 | 70.21 | 83.82 | 90.75 |
| | GRPO | 17.99 | 62.85 | 73.83 | 86.68 | 92.52 |

**Backbone models.** Across both Qwen2.5-VL-7B [24] and InternVL3-8B [33], GRPO consistently yields clear improvements over SFT at all geographical levels (see Table 4), confirming the robustness of the training framework. Nevertheless, the absolute performance is influenced by the backbone itself, with Qwen2.5-VL-7B [24] achieving higher post-GRPO accuracy than InternVL3-8B [33]. These results indicate that GRPO provides stable relative gains across architectures, while the final performance ceiling is determined by backbone capacity and pretraining quality.

Table 5: Ablation on data curation with Qwen2.5-VL-7B [24] backbone.

| Curation Setting | Training Strategy | Street 1km | City 25km | Region 200km | Country 750km | Continent 2500km |
|---|---|---|---|---|---|---|
| | | | | *MP16-Reason*-Test (% @ km) | | |
| Baseline | - | 15.42 | 52.72 | 62.86 | 75.11 | 83.47 |
| Random sampling | SFT | 15.23 | 52.00 | 64.56 | 78.17 | 85.23 |
| | GRPO | 17.26 | 59.22 | 71.80 | 84.73 | 91.26 |
| Single-source validation | SFT | 15.22 | 52.47 | 65.09 | 78.79 | 86.15 |
| | GRPO | 17.37 | 59.45 | 71.88 | 84.74 | 91.24 |
| Full multi-source validation | SFT | 16.38 | 56.76 | 70.21 | 83.82 | 90.75 |
| | GRPO | 17.99 | 62.85 | 73.83 | 86.68 | 92.52 |

**Distillation datasets.** To evaluate the contribution of our validation steps in data curation (*MP16-Reason*-Train), we compare models trained on different curation settings, all standardized to 33K samples but obtained through different filtering strategies, such as random sampling or validation by only a single LVLM (InternVL3-78B [33]). As shown in Table 5, these ablated settings lead to noticeable performance drops compared with the full *MP16-Reason*-Train, confirming the importance of comprehensive validation in constructing a high-quality dataset. In addition, GRPO consistently outperforms SFT across all settings, further highlighting the effectiveness of reinforcement learning in leveraging reasoning-driven supervision.

# 5 Discussion

**Toward Fine-Grained Geo-localization: Limits of Pure Reasoning.** While our reasoning-driven framework achieves strong performance at the country and city levels, its effectiveness diminishes when tasked with fine-grained, coordinate-level localization. This limitation originates from the inherent nature of the reasoning process: predictions are based on high-level semantic cues such as language, architectural style, or vegetation, which often lack the spatial specificity required to differentiate between closely situated locations. For example, multiple European cities may share similar visual patterns, such as Mediterranean-style architecture, the presence of European Union flags, or public signage in English, which makes it difficult for the model to resolve fine-grained geographic ambiguities through reasoning alone. In such cases, even accurate reasoning can only narrow down a broad region but cannot pinpoint an exact location. This highlights a key challenge in reasoning-driven geo-localization: the lack of precise visual-geographic anchoring. To overcome this limitation, future work may explore hybrid approaches that combine reasoning to constrain the candidate region, followed by local feature-based retrieval within that region to achieve coordinate-level precision.

**Beyond Scale Alone: Data Efficiency in Reasoning-driven Training.** Our experiments show that training *GLOBE* on just 33K high-quality, reasoning-oriented samples (*MP16-Reason*) achieves performance comparable to, and sometimes exceeding, models trained on millions of generic image-text pairs. This highlights that for reasoning-driven tasks, targeted supervision can be more effective than sheer data scale. Our results suggest that aligning supervision with task-specific reasoning offers a more data-efficient path forward for LVLM training.

**Beyond Geo-localization: GRPO for Reasoning-driven LVLM Tasks.** Our findings suggest that GRPO, as a training paradigm, is particularly well-suited for reasoning-driven objectives in LVLMs. Unlike SFT, which often treats outputs as isolated targets, GRPO directly optimizes the relative quality of outputs through scalar reward signals. This form of supervision allows GRPO to guide complex reasoning behaviors in a more structured and interpretable manner than traditional training objectives. While our work focuses on geo-localization, we believe the GRPO paradigm can be readily extended to other multimodal reasoning tasks, such as visual question answering and multimodal chain-of-thought generation.

# 6 Conclusion

In this paper, we present a novel reasoning-driven pipeline for image geo-localization by leveraging LVLMs. To address the limitations of existing datasets and training paradigms, we introduce *MP16-Reason*, a high-quality dataset constructed from diverse social media images and enriched with automatically distilled localizability labels and reasoning trajectories. Building upon this dataset, we propose *GLOBE*, an LVLM trained via GRPO-based reinforcement learning, which jointly improves three core aspects of geo-localization: localizability assessment, visual-cue reasoning, and geo-location recognition. In contrast to SFT, our GRPO-based training framework directly optimizes reasoning quality through structured reward signals, leading to substantial gains in both interpretability and localization accuracy. Empirical results show that *GLOBE*, using only 33K data, achieves performance comparable to or better than state-of-the-art methods trained on millions of samples.

## Acknowledgements

The authors would like to thank the anonymous reviewers for their valuable comments. This work is funded by National Natural Science Foundation of China Grant (72371217, 62402414), the Guangzhou Industrial Informatics and Intelligence Key Laboratory No. 2024A03J0628, the Nansha Key Area Science and Technology Project No. 2023ZD003, and Project No. 2021JC02X191, and the Guangdong Basic and Applied Basic Research Foundation No. 2025A1515011994.

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

# A    Appendix

## A.1    Implementation Details

### A.1.1    Prompt Template

To ensure fairness and consistency in both training and evaluation, we standardize the input format for all LVLMs. Specifically, images are fed into the model along with a unified prompt template. This prompt is designed to guide the model toward reasoning about geographic cues in a consistent manner across different stages and model variants. During both fine-tuning and testing, we use the same prompt format to minimize prompt-induced variance and allow for fair comparison of model performance.

---

**Prompt Template**

You are a geolocation expert. You are participating in a geolocation challenge. Based on the provided image:
1. Carefully analyze the image for clues about its location (architecture, signage, vegetation, terrain, etc.)
2. Think step-by-step about what country, and city this is likely to be in and why

Your final answer includes these two lines somewhere in your response:
country: [country name]
city: [city name]

You MUST output the thinking process in <think> </think> and give the answer in <answer> </answer> tags.

---

### A.1.2    Training Data Example

The example below illustrates a training sample from *MP16-Reason*. Each instance contains the image path, a reasoning trajectory generated by vision-language models, ground-truth city and country labels, and a set of grounded entities extracted from the reasoning text.

---

**Training Data Example**

**img_path:** "3f_e4_302010632.jpg"
**reason:** "The image shows a large stadium filled with spectators wearing red clothing, indicating a strong team spirit and possibly a sports event.
There is a scoreboard displaying 'Ohio State' and 'Michigan', which are well-known college football teams from the United States. This suggests that the event is an American college football game.
The presence of cheerleaders, a marching band, and the large crowd further supports the idea that this is a major sporting event in the United States.
The architecture of the stadium, including the design of the scoreboard and the layout of the seating, is consistent with large American college football stadiums.
Given the reference to 'Ohio State', it is highly likely that this event is taking place in Ohio, specifically at Ohio State University's home stadium, which is located in Columbus."
**city:** "Columbus"
**country:** "United States"
**entities:** [{"text": "stadium", "type": "ARCH"}, {"text": "scoreboard", "type": "SIGN"}, {"text": "seating", "type": "ARCH"}]

---

### A.1.3    Hyper-parameter Settings

We summarize the key hyper-parameters used in training *GLOBE* in Table 6. These settings are selected based on standard practices in fine-tuning large vision-language models and further adjusted through preliminary ablation studies on a held-out validation set. Unless otherwise specified, all experiments are conducted using the same configuration to ensure comparability and reproducibility.

Table 6: The hyper-parameter settings of the proposed *GLOBE*.

| Hyper Params | Value |
|---|---|
| Learning Rate | 1e-6 |
| Total Batch Size | 16 |
| Weight Decay | 0.1 |
| Warmup Ratio | 0.01 |
| Optimizer | AdamW |
| Adam Beta1 | 0.9 |
| Adam Beta2 | 0.95 |
| LR Scheduler | cosine |
| Model Max Length | 8192 |

## A.2  Experimental Results

### A.2.1  The performance of *GLOBE* across different conditions

Table 7 presents geo-localization performance across three scene types (indoor, nature, and urban). We compare *GLOBE*-7B with Qwen2.5-VL-7B [24] and GeoReasoner-7B [28]. Across all conditions, *GLOBE* delivers consistently higher accuracy at every geographical level, demonstrating robust performance in diverse visual environments.

Table 7: Geo-localization performance comparison across different conditions.

| Method | MP16-*Reason*-Test (% @ km) | | | | |
|---|---|---|---|---|---|
| | Street 1km | City 25km | Region 200km | Country 750km | Continent 2500km |
| **I. Indoor Scene** | | | | | |
| Qwen2.5-VL-7B [24] | 12.50 | 46.95 | 55.30 | 69.99 | 81.49 |
| GeoReasoner-7B [28] | 12.57 | 35.93 | 48.50 | 65.87 | 79.04 |
| *GLOBE*-**7B (Ours)** | 17.65 | 57.35 | 64.71 | 80.88 | 91.18 |
| **II. Nature Scene** | | | | | |
| Qwen2.5-VL-7B [24] | 8.61 | 42.77 | 60.07 | 72.62 | 80.68 |
| GeoReasoner-7B [28] | 5.10 | 35.71 | 48.98 | 67.35 | 78.57 |
| *GLOBE*-**7B (Ours)** | 13.95 | 55.81 | 81.40 | 90.70 | 97.67 |
| **III. Urban Scene** | | | | | |
| Qwen2.5-VL-7B [24] | 16.95 | 55.32 | 65.00 | 76.63 | 84.29 |
| GeoReasoner-7B [28] | 10.18 | 42.23 | 51.86 | 68.78 | 80.19 |
| *GLOBE*-**7B (Ours)** | 18.61 | 64.98 | 74.76 | 87.38 | 92.11 |

Best results are in blue.

Table 8: Geo-localization performance comparison on OSV-5M [83] (mini-3K).

| Method | OSV-5M [83] (mini-3K) (% @ km) | | | | |
|---|---|---|---|---|---|
| | Street 1km | City 25km | Region 200km | Country 750km | Continent 2500km |
| ISNs [9] | 0.00 | 1.07 | 6.77 | 22.04 | 44.01 |
| GeoCLIP [15] | 0.07 | 1.57 | 13.87 | 44.51 | 73.26 |
| Qwen2.5-VL-7B [24] | 0.00 | 0.87 | 5.14 | 19.81 | 40.55 |
| InternVL3-8B [33] | 0.00 | 0.73 | 5.27 | 19.81 | 44.01 |
| *GLOBE*-**7B (Ours)** | 0.00 | 1.87 | 14.04 | 45.01 | 74.16 |

Best results are in blue.

### A.2.2 The performance of *GLOBE* on street-view images

Table 8 reports geo-localization performance on OSV-5M [83] (mini-3K), a benchmark consisting exclusively of street-view imagery. We compare *GLOBE*-7B against ISNs [9], GeoCLIP [15], and backbone-matched LVLMs such as Qwen2.5-VL-7B [24] and InternVL3-8B [33]. *GLOBE* achieves the best results at the city, region, country, and continent levels, demonstrating strong generalization to a domain outside its training distribution.

### A.2.3 Failure cases

We categorize failure cases into two types: *Error Reasoning* and *Right Reasoning*, with representative examples illustrated in Table 9. *Error Reasoning* refers to cases where the model generates incorrect or irrelevant reasoning steps that misinterpret visual cues or contextual evidence. In contrast, *Right Reasoning* describes cases where the reasoning process is logically sound and visually grounded, yet the final prediction is incorrect (often due to dataset bias or overrepresentation of certain locations). Our analysis reveals two common patterns: ① visually similar features (e.g., domes, arches) leading to incorrect landmark attribution, and ② correct reasoning that is nevertheless biased toward locations more frequently represented in the training data.

Table 9: Representative failure cases.

| Type | IMG_ID | Ground Truth | Reasoning | Prediction |
|------|--------|--------------|-----------|------------|
| ① | eb_80_511397613.jpg | Etterbeek, Belgium | ***Terrain and Urban Setting***: *The grand scale of the architecture with large columns and arches resembles the Louvre Pyramid or the Arc de Triomphe area.* | *Paris, France* |
| ② | 98_80_2913796353.jpg | Şehitkamil, Turkey | ***Architecture***: *The minaret has a cylindrical structure with multiple levels featuring ornamental details like intricate carvings, wooden balconies with latticework, and stone masonry. The domed roof with a pointed tip and the presence of a crescent moon finial suggests Ottoman or Islamic architectural style.* | *Baku, Azerbaijan* |

### A.2.4 Inference efficiency

Table 10 reports the latency, throughput, and streaming metrics, with all models tested under the same hardware (single H20 GPU) and software settings (e.g., PyTorch, vLLM) as *GLOBE*. While *GLOBE* delivers stronger accuracy, it incurs higher average latency and lower throughput, reflecting the additional reasoning time required by larger LVLMs.

Table 10: Inference efficiency comparison between different baseline methods.

| Model | Concurrency | Streaming | Avg. Latency (s) | Throughput (QPS) | TTFT (ms) | TPOT (ms) |
|-------|-------------|-----------|------------------|------------------|-----------|-----------|
| GeoCLIP [15] | 1 | - | 0.1364 | 7.3313 | - | - |
| RFM-YFCC [49] | 1 | - | 0.5852 | 1.7088 | - | - |
| Qwen2.5-VL-7B [24] | 1 | No | 2.9368 | 0.3405 | - | - |
| InternVL3-8B [33] | 1 | No | 3.4886 | 0.2866 | - | - |
| *GLOBE* (InternVL3-8B [33]) | 1 | No | 4.8742 | 0.2051 | - | - |
| *GLOBE* (InternVL3-8B [33]) | 1 | Yes | 4.8597 | 0.2057 | 64.23 | 11.70 |
| *GLOBE* (Qwen2.5-VL-7B [24]) | 1 | No | 4.5684 | 0.2188 | - | - |
| *GLOBE* (Qwen2.5-VL-7B [24]) | 1 | Yes | 4.5628 | 0.2191 | 34.67 | 11.72 |
| *GLOBE* (Qwen2.5-VL-7B [24]) | 8 | No | 5.1415 | 1.5045 | - | - |
| *GLOBE* (Qwen2.5-VL-7B [24]) | 8 | Yes | 5.1589 | 1.4990 | 82.16 | 13.67 |
| *GLOBE* (Qwen2.5-VL-7B [24]) | 32 | No | 6.2479 | 4.5370 | - | - |
| *GLOBE* (Qwen2.5-VL-7B [24]) | 32 | Yes | 6.2923 | 4.5602 | 161.40 | 17.12 |

### A.2.5 Qualitative results

Figure 5 shows that *GLOBE* produces reasoning trajectories with improved coherence and interpretability. In particular, the model engages in structured reasoning to derive geo-location predictions, systematically incorporating diverse geographic cues such as architectural style, signage, vegetation, and other contextually informative elements.

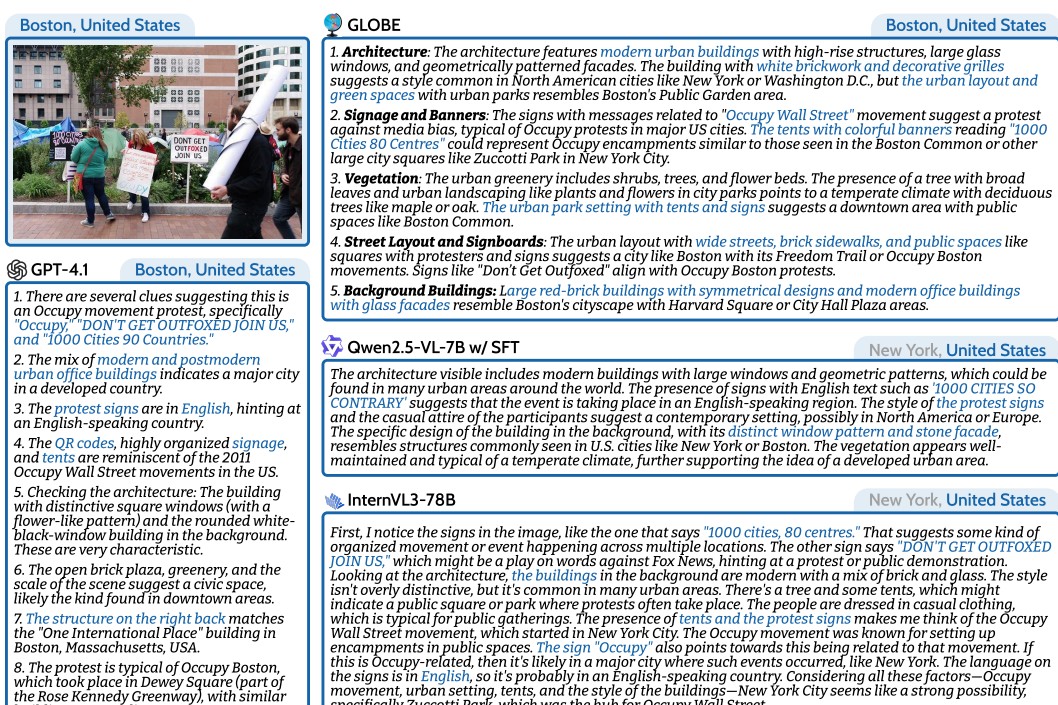

Figure 5: Reasoning comparison of four different models (GPT-4.1 [86], *GLOBE*, Qwen2.5-VL-7B [24] with SFT, and InternVL3-78B [33]) on the same input image. Reliable visual cues identified by the models are marked in text.

### A.2.6 Hyperparameters $\lambda_1$, $\lambda_2$ and $\lambda_3$ in GRPO

The weights $\lambda_1$, $\lambda_2$, and $\lambda_3$ are fixed during training and were initially set to 0.2, 0.5, and 1, respectively. Since the primary objective is accurate prediction of city- and country-level locations, $\lambda_3$ (which directly supervises geo-localization accuracy) was assigned the highest weight. To mitigate hallucinations in the reasoning trajectory, $\lambda_2$ (consistency) was given a relatively high weight. In contrast, $\lambda_1$ (localizability), a binary score that evaluates whether the reasoning is geographically grounded, was assigned a smaller value to act as auxiliary regularization. As shown in Table 11, different weight combinations were tested, and the proposed setting (0.2, 0.5, 1) yielded the best performance.

Table 11: Performance comparison of weight selection ($\lambda_1$, $\lambda_2$ and $\lambda_3$) for the GRPO framework.

| $\lambda_1$ | $\lambda_2$ | $\lambda_3$ | Street 1km | City 25km | Region 200km | Country 750km | Continent 2500km |
|---|---|---|---|---|---|---|---|
| | | | \multicolumn *MP16-Reason*-Test (% @ km) | | | | |

| $\lambda_1$ | $\lambda_2$ | $\lambda_3$ | Street 1km | City 25km | Region 200km | Country 750km | Continent 2500km |
|---|---|---|---|---|---|---|---|
| 1.0 | 0.5 | 0.2 | 17.63 | 59.96 | 72.11 | 84.87 | 91.55 |
| 1.0 | 1.0 | 1.0 | 17.67 | 59.94 | 71.83 | 84.80 | 91.20 |
| **0.2** | **0.5** | **1.0** | **17.99** | **62.85** | **73.83** | **86.68** | **92.52** |

The chosen configurations of $\lambda_1$, $\lambda_2$, and $\lambda_3$ are marked in **bold**.

### A.2.7 Analysis of Reward Trajectories

The training dynamics of the three reward signals show generally upward trends (see Figure 6). The Localizability Reward steadily increases and gradually plateaus, showing consistent improvement in identifying informative inputs. Thanks to the pre-filtering of training data, both the Localizability Reward and the Geo-localization Accuracy Reward start with relatively high values and maintain strong performance even in the early training steps. The Visual Grounding Consistency Reward rises quickly during the initial stage before stabilizing, indicating that the model rapidly learns to associate

visual entities with their corresponding locations. Overall, the trends of all three rewards suggest stable and effective learning throughout training.

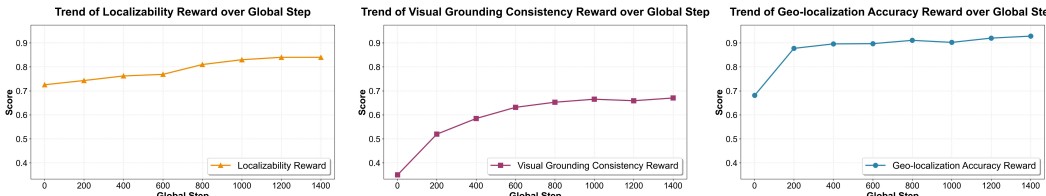

Figure 6: Training dynamics of three rewards over global steps. From left to right: (a) Localizability Reward, (b) Visual Grounding Consistency Reward, and (c) Geo-localization Accuracy Reward. Each curve shows how the corresponding reward evolves as training progresses.

## A.3  Limitation

While our reasoning-based framework performs well at country and city levels, its accuracy declines in fine-grained, coordinate-level geo-localization. This is due to the abstract nature of reasoning, which relies on high-level semantic cues (e.g., architecture, language, vegetation) that often lack the spatial precision needed to distinguish between visually similar, nearby locations. As a result, even correct reasoning may only localize to a broad region. Future work could address this by combining reasoning with local feature-based retrieval to improve fine-grained accuracy.

## A.4  Broader Impacts

This work introduces a reasoning-oriented geo-localization framework that leverages diverse social media imagery and bi-objective optimization to enhance the reasoning capabilities of large vision-language models. While this approach improves interpretability and performance in complex visual scenes, it also raises privacy and misuse concerns. The ability to infer precise locations from user-shared images may lead to unauthorized tracking, surveillance, or profiling, especially if deployed at scale without appropriate safeguards.

To mitigate these risks, we recommend restricting access to the model via gated APIs, incorporating uncertainty estimation in predictions, and clearly documenting limitations and intended use cases. Special care should be taken when applying the method to user-generated content, including adherence to data licenses and privacy-preserving practices. Responsible deployment will be essential to ensure the benefits of improved geo-reasoning do not come at the cost of societal harm.

