# OpenReview forum: "Recognition through Reasoning: Reinforcing Image Geo-localization with Large Vision-Language Models"
_NeurIPS.cc/2025/Conference — NeurIPS 2025 poster_

### Official Review · Reviewer_WR7w · 2025-06-09

**Clarity:** 3
**Significance:** 3
**Originality:** 3
**Rating:** 4
**Confidence:** 5

**Summary:**

This paper presents GLOBE, an open-source model for worldwide image geolocalization. The authors first augmented the existing open-source dataset MP16 by adding explicit reasoning trajectories data to construct MP16-reason. Then, they fine-tuned the open-source qwen2.5-vl model using the GRPO reinforcement learning algorithm. Experiments are conducted on their curated dataset and IM2GPS3K.

**Questions:**

Please refer to the weaknesses.

If authors can finish the experiments I mentioned in the weaknesses and provide a reasonable explanation for the issues I raised, I will raise the score.

**Ethical Concerns:**

["NO or VERY MINOR ethics concerns only"]

**Final Justification:**

The authors' rebuttal has addressed most of my concerns. For the follow-up questions Q1 and Q3, I still have some minor confusion.
However, overall, I believe this paper offers a new perspective for image geolocalization by incorporating a novel reasoning approach.
Therefore, I have updated my rating to Borderline accept.

**Limitations:**

Yes

**Paper Formatting Concerns:**

No major formatting issues.

**Quality:**

2

**Strengths And Weaknesses:**

Strengths:
1. The code is open-sourced, facilitating further research.
2. The related work is very comprehensive.

Weaknesses:
1. When building MP16-Reason, there are opportunities that the reasoning trajectories are wrong but lead to correct predictions by coincidence [1]. Have you thought about these situations?
2. Lines 149-152, the authors mention how they validate with the semantic segmentation models, but this part is really short and hard to understand.
3. Line 174, why is this reward designed this way? Is it meant that all elements in $v_i$ must be mentioned in $\hat{r_i}$? This seems too strict because not all entities are useful for geolocation. If it only requires matching any one of them, then the standard feels too loose. In addition, the description here is not very clear.
4. For Image Geolocalization task's datasets, for example, the IM2GPS3K or YFCC4K, they are usually evaluated with 5-level metrics (from 1km level to 2500km level). But authors only consider the middle three levels.
5. Limited experiments, some experiments are missing. The authors only conduct the overall experiments and ablation study; no further experiments are conducted.
* For example, what's the impact of $\lambda_1, \lambda_2, \lambda_3$ on the performance?
* How does the performance change when using different backbone models?
* How the performance changes if we do not use 3 validation methods in dataset curation?
6. The structure of the experiment section is really confusing. Sections 4.2.1 and 4.2.2, as well as Table 1 and Table 2, are mixed together in a way that is very difficult to understand and read.

Minors:
1. It would be better to clarify the model size used in GLOBE in Table 1.
2. Figure 5 is more like a case study. It would be better to open a new section for this figure.

[1] Guo, Daya, et al. "Deepseek-r1: Incentivizing reasoning capability in llms via reinforcement learning." arXiv preprint arXiv:2501.12948 (2025).

---

> ### Author Rebuttal · Authors · 2025-07-30
>
> Thank you for the thoughtful and detailed feedback. We sincerely appreciate your recognition of the $\textbf{open-sourced code}$ and the $\textbf{comprehensiveness of our related work}$, as well as the time and effort you put into reviewing our paper. Below, we address your concerns point by point to further clarify and strengthen our contributions.
>
> > ### 1. The explanation of potential correct-by-coincidence reasoning traces (W1)
>
> A1: While some distilled reasoning traces may contain flawed logic that coincidentally yields correct predictions, we emphasize that $\textbf{these raw traces are not directly used in GRPO training}$.
> Instead, we extract key entities (e.g., architecture, plant) and cross-validate them against predictions from a semantic segmentation model. Only entities consistent with visual evidence are retained for supervision, significantly reducing the risk of propagating spurious reasoning.
> Since SFT directly learns from the full synthetic reasoning traces, it is more vulnerable to coincidental correctness, where flawed reasoning leads to correct predictions. In contrast, our GRPO-based approach relies on verified, structured signals (e.g., entity-level consistency), making it more robust to such artifacts. This distinction is reflected in Table 2 of the main manuscript (Row 2 vs. Rows 3–8), where GRPO variants consistently outperform standard SFT.
>
> > ### 2. The explanation of validation with the semantic segmentation models (W2)
>
> A2: We provide additional clarification on how semantic segmentation is used to validate reasoning traces.
> As described in Lines 149–152, we apply MaskFormer [1], a state-of-the-art panoptic segmentation model, to each image in our dataset to obtain pixel-wise semantic predictions across 150 common categories, including sky, building, tree, road, vegetation, car, and others. This provides a dense, localized understanding of the visual content.
> Crucially, this segmentation output is used to verify the factual consistency of entities mentioned in the distilled reasoning traces. For example, if the reasoning claims "the presence of snow-capped mountains suggests high altitude," we check whether mountain and snow regions are indeed present in the corresponding image regions. Similarly, if a trace references "urban infrastructure" or "dense road networks," we validate whether buildings and roads are detected with high confidence and spatial coverage.
> This cross-modal validation ensures that only reasoning paths grounded in actual visual content are preserved, reducing the risk of propagating hallucinated or coincidentally correct logic. We acknowledge that this part was briefly described in the original submission, and we will expand it in the revised manuscript with a clearer explanation and an example in the appendix.
>
>
> > ### 3. The explanation of Visual Grounding Consistency Reward (W3)
>
> A3: The reward design in Line 174 is based on the entity-level consistency between the reasoning trace $\hat{r_i}$ and the visual content of the image, as verified by semantic segmentation (see A2).
> Specifically, we extract the intersection of entities present in both the reasoning trace and the MaskFormer-predicted segmentation map. This does not require all detected visual entities to be mentioned (which would indeed be overly strict), nor does it consider a match valid if any single entity aligns (which would be too lenient). Instead, the reward reflects the proportion of mentioned entities that are visually grounded, normalized by the total number of relevant entities in the trace.
> As shown in Table 2 of the main manuscript (Row 7 vs. Row 2), this consistency-based reward leads to significant performance gains compared to standard SFT, demonstrating its effectiveness in guiding geolocation-capable reasoning. We will revise the text to include a formal definition of the reward and an illustrative example in the appendix.
>
> >  ### 4. The explanation of only considering the middle three levels (W4)
>
> A4: We acknowledge that standard benchmarks are typically evaluated using five distance-based accuracy levels (1km, 25km, 200km, 750km, 2500km). In our work, however, the primary focus is on reasoning-based geolocation, where the model generates semantic reasoning traces rather than predicting precise coordinates. Given this design, the localization resolution is inherently limited to coarser granularity, making performance at the 1km level less reflective of the model’s reasoning capability.
> To maintain consistency with our method’s scope, we initially reported results on the middle three levels (25km, 200km, 750km) in the main text, which better align with city- and country-level predictions. As shown in Table 1 (updated), we now include results across all five levels.
>
> ### Table 1. Performance of GLOBE on IMG2GPS3k
> | Method | Street (1km) | City (25km) | Region (200km) | Country (750km) | Continent (2500km) |
> | -------| -------- | -------- | --------- | -------- | -------- |
> | ISNs | 10.50 | 28.00 | 36.60 | 49.70 | 66.00 |
> | GeoCLIP | 14.11 | 34.47 | 50.65 | 69.67 | 83.82 |
> | Qwen2.5-VL-7B | 8.58 | 32.53 | 43.11 | 58.93 | 72.37 |
> | GLOBE (ours) | 9.84 | 40.18 | 56.19 | 71.45 | 82.38 |
>
> As expected, performance at the 1km level is lower, which is consistent with the fact that our approach does not aim for street-level precision.
>
> > ### 5. Additional experiments for the proposed method (W5)
>
> A5:  We agree that deeper ablation and sensitivity analyses are important for understanding the robustness and generalizability of our method. In response, we have conducted three additional sets of experiments, which will be included in the revised manuscript:
>
> $\textbf{a. Sensitivity to reward weights ($λ_1$, $λ_2$, $λ_3$)}$: The weights  λ₁, λ₂, and λ₃ are fixed during training and were initially set to 0.2, 0.5, and 1, respectively. Since the primary objective is to accurately predict the city and country, we assigned the highest weight to λ₃, which directly supervises geolocation accuracy. Consistency is important for reducing hallucinations in the reasoning trajectory; thus, we assigned a relatively high weight to λ₂. Locatability, being a binary score that evaluates whether the image and its reasoning are geographically grounded, was assigned a smaller weight (λ₁) to provide auxiliary regularization. As shown in Table 2, we evaluate different weight combinations in the GRPO objective. The results confirm that our chosen setting (0.2,0.5,1) achieves the best performance.
>
> ### Table 2. Performance comparison of weight selection ( λ₁, λ₂, and λ₃) for the GRPO framework on MP16-Reason-Test
> | GRPO | Street (1km) | City (25km) | Region (200km) | Country (750km) | Continent (2500km) |
> | ------- | -------- | -------- | -------- |  -------- | -------- |
> | λ₁, λ₂,  λ₃ = 0.2, 0.5, 1 (proposed setting) | 17.99 | 62.85 | 73.83 | 86.68 | 92.52 |
> | λ₁, λ₂,  λ₃ = 1, 1, 1 | 17.67 | 59.94 | 71.83 | 84.80 | 91.20 |
> | λ₁, λ₂,  λ₃ = 1, 0.5, 0.2 | 17.63 | 59.96 | 72.11 | 84.87 | 91.55 |
>
> $\textbf{b. Impact of backbone architecture}$: Table 3 compares performance across different vision-language model backbones (e.g., Qwen2.5-VL-7B vs. InternVL3-8B). The consistent gains of GRPO over SFT across backbones demonstrate the generalizability of our training framework.
>
> ### Table 3. Performance comparison using different backbones on MP16-Reason-Test
> | Backbone | Training Strategy | Street (1km) | City (25km) | Region (200km) | Country (750km) | Continent (2500km) |
> | ------- | -------- | -------- | --------- | -------- | -------- | -------- |
> | Qwen2.5-VL-7B | Baseline | 15.42 | 52.72 | 62.86 | 75.11 | 83.47|
> | Qwen2.5-VL-7B | SFT | 16.38 | 56.76 | 70.21 | 83.82 | 90.75|
> | Qwen2.5-VL-7B | GRPO | 17.99 | 62.85 | 73.83 | 86.68 | 92.52|
> | InternVL3-8B | Baseline |12.01 | 44.17 | 55.66 | 75.36 | 86.98|
> | InternVL3-8B | SFT |12.41 | 44.68 | 56.37 | 75.20 | 86.32|
> | InternVL3-8B | GRPO |17.47 | 60.09 | 72.41 | 85.02 | 91.92|
>
> $\textbf{c. Ablation on dataset curation}$: To evaluate the contribution of our validation steps in data curation, we are conducting experiments using ablated training sets (e.g., randomly sampled data, or data with only one LVLM validation). Table 4 shows clear performance drops when any of the validation steps are removed, confirming their importance in building a high-quality dataset.
>
> ### Table 4. Performance comparison using different Training datasets on MP16-Reason-Test, with Qwen2.5-VL-7B
>
> | Training Dataset | Training Strategy | Street (1km) | City (25km) | Region (200km) | Country (750km) | Continent (2500km) |
> | -------- | -------- | --------- | -------- | -------- | -------- | -------- |
> | Baseline | - | 15.42 | 52.72 | 62.86 | 75.11 | 83.47|
> | Random | SFT | 15.23 | 52.00 | 64.56 | 78.17 | 85.23 |
> | Random | GRPO | 17.26 | 59.22 | 71.80 | 84.73 | 91.26 |
> | Only InternVL3-78B distill | SFT | 15.22 | 52.47 | 65.09 | 78.79 | 86.15 |
> | Only InternVL3-78B distill | GRPO | 17.37 | 59.45 | 71.88 | 84.74 | 91.24 |
> | MP16-Reason-Train | SFT | 16.38 | 56.76 | 70.21 | 83.82 | 90.75|
> | MP16-Reason-Train | GRPO | 17.99 | 62.85 | 73.83 | 86.68 | 92.52|
>
> > ### 6. The writing suggestions for the experiment section. (W6)
>
> A6:  We thank the reviewer for the feedback. We agree that the current structure may be confusing and will revise the experimental section to clearly separate main results (Section 4.2.1, Table 1) from ablation studies (Section 4.2.2, Table 2). We will clarify the purpose of each table and improve the flow in the revised manuscript.
>
> > ### 7. Some minors (W7)
>
> A7: We thank the reviewer for the suggestions. We will clarify the model size used in GLOBE in Table 1 and move the case study in Figure 5 to a dedicated subsection (e.g., "Qualitative Analysis" or "Case Study") for better organization.
>
> [1] Per-Pixel Classification is Not All You Need for Semantic Segmentation, NeurIPS 2021

---

> > ### Comment · Reviewer_WR7w · 2025-08-01
> >
> > Thank you very much to the authors for the response. The additional experiments undoubtedly strengthen the paper. I acknowledge that the authors’ reply has addressed some of my concerns, and I commit to raising my score.
> >
> > To further convince me, I would like to discuss the following three questions:
> > 1. From the examples provided by the authors, the model outputs city and country names. I am just curious, how did the authors obtain the corresponding GPS coordinates for city, region, and country names? Additionally, in Table 1, GLOBE shows a slight drop compared to GeoCLIP at the continent level, which seems somewhat counterintuitive. Do the authors have any insights on this result or potential solutions?
> > 2. Thank you for providing additional results regarding the reward weights. Here I would like to raise a point for discussion: since the three rewards are all within the [0, 1] range, using weights like (0.2, 0.5, 1) versus (1, 1, 1) may introduce further confounding factors. For instance, when using (1, 1, 1), the absolute differences among reward values might be exaggerated simply due to scaling. Would it be more reasonable to normalize the sum of the lambdas to a constant across different settings? I do not have a definitive solution here, just hoping to hear the authors’ thoughts.
> > 3. Since the proposed method involves a reasoning-based model, efficiency is also a key consideration. May I ask how long it takes, approximately, to process one image from the Im2GPS3k dataset using Qwen2.5-VL-7B as the backbone? This is an important factor when considering real-world deployment.
> >
> > Please feel free to respond. I will further consider my final rating based on the authors’ reply. And thanks again for your hard work in rebuttal.

---

> > > ### Author Response · Authors · 2025-08-05
> > >
> > > We sincerely thank the reviewer for the follow-up and for considering increasing the score. We are pleased to address the three additional questions below:
> > >
> > > > ### Q1: Clarification on obtaining GPS coordinates
> > >
> > > To convert the predicted location names into GPS coordinates, we concatenate the predicted city and country names into a single string and query the Microsoft Azure Maps tools, which returns a representative coordinate (e.g., the geographic center of the region).
> > >
> > > This design also explains the drop in continent-level accuracy in Table 1. Even when the predicted location name is correct, the queried coordinates may deviate from the ground truth, introducing spatial errors that affect coarse-grained evaluations.
> > >
> > > Regarding potential solutions, future work could adopt a two-step strategy: first, predict coarse-grained semantic locations (e.g., city or country), then refine to GPS coordinates. This hierarchical approach may enhance accuracy across spatial granularities while reducing reliance on external geocoding APIs.
> > >
> > > > ### Q2: The discussion of λ
> > >
> > > The reason we use different $\lambda$ values is to investigate the impact of each reward component on the task. As we show below, it is the **relative values** of the weights that ultimately affect the learning process, not their absolute magnitudes.
> > >
> > > In our experiments, we tested three settings: $[0.2, 0.5, 1]$, $[1, 0.5, 0.2]$, and $[1, 1, 1]$. The first two have a total weight sum of $1.7$, while the last has a sum of $3$. However, due to the normalization step in GRPO [1], the setting $[1,1,1]$ is effectively equivalent to $[1.7/3, 1.7/3, 1.7/3]$, because both represent the same relative proportions (i.e., equal weighting across components).
> > >
> > > To generalize this point, consider a scaled version of any base weight configuration:
> > >
> > > $\lambda'_1 = K \cdot \lambda_1$,  $\lambda'_2 = K \cdot \lambda_2$, $\lambda'_3 = K \cdot \lambda_3$, where $K > 0$ is a constant scaling factor.
> > >
> > > We now substitute these into the reward computation in GRPO:
> > >
> > > $r_i = \lambda'_1 \cdot r_i^{\text{loc}} + \lambda'_2 \cdot r_i^{\text{vis}} + \lambda'_3 \cdot r_i^{\text{geo}} = K \left( \lambda_1 \cdot r_i^{\text{loc}} + \lambda_2 \cdot r_i^{\text{vis}} + \lambda_3 \cdot r_i^{\text{geo}} \right)$
> > >
> > > Next, the reward is normalized across the batch:
> > > $\hat{r}_i = \frac{r_i - \mu_r}{\sigma_r}, \quad \text{for } i \in [1, G]$
> > >
> > > where $\mu_r = \frac{1}{G} \sum_{i=1}^{G} r_i$, $\sigma_r = \sqrt{\frac{1}{G} \sum_{i=1}^{G} (r_i - \mu_r)^2}$ and $G$ is the number of sampled outputs.
> > >
> > > Now substitute $r_i = K \cdot r_i^{\text{raw}}$, where $r_i^{\text{raw}} = \lambda_1 \cdot r_i^{\text{loc}} + \lambda_2 \cdot r_i^{\text{vis}} + \lambda_3 \cdot r_i^{\text{geo}}$:
> > >
> > > Since $\hat{r}_i = (r_i - \mu_r)/\sigma_r$, and $r_i = K \cdot r_i^{\text{raw}}$, we have $\mu_r = K \cdot \mu_r^{\text{raw}}$, $\sigma_r = K \cdot \sigma_r^{\text{raw}}$, so:
> > >
> > > $$
> > > \hat{r}_i = \frac{K r_i^{\text{raw}} - K \mu_r^{\text{raw}}}{K \sigma_r^{\text{raw}}} = \frac{r_i^{\text{raw}} - \mu_r^{\text{raw}}}{\sigma_r^{\text{raw}}}
> > > $$
> > >
> > > As we can see, the scaling factor $K$ cancels out completely. The final normalized reward $\hat{r}_i$ is independent of $K$.
> > >
> > > Therefore, what truly matters is the **relative ratio** among the $\lambda_j$ values, not their absolute scale. Our experimental design is thus valid and free from confounding due to weight magnitude differences.
> > >
> > > We hope this addresses the reviewer’s concern and clarifies why our current setup is mathematically sound and robust in practice.
> > >
> > > > ### Q3: Inference Efficiency
> > >
> > > Our GLOBE (Qwen2.5-VL-7B backbone) is deployed via vLLM (v0.8.0) in online serving mode using 80 GB memory on a single H20 GPU. Table 5 reports average latency and throughput on the Im2GPS3k test set under different concurrency levels, covering both visual encoding and reasoning generation. TTFT (Time to First Token) and TPOT (Time Per Output Token) are measured in milliseconds in streaming output mode to reflect real-world usage.
> > > The runtime is comparable to other 7B-level LVLMs.
> > >
> > > ### Table 5. Inference efficiency of GLOBE (Qwen2.5-VL-7B backbone) on the Im2GPS3k test set under varying concurrency levels
> > > | Concurrency | Streaming | Avg. Latency (s) | Throughput (QPS) | TTFT (ms) | TPOT (ms) |
> > > |--|--|--|--|--|--|
> > > | 1  | No | 4.5684 | 0.2188 | - | - |
> > > | 1  | Yes | 4.5628 | 0.2191| 34.67 | 11.72 |
> > > | 8  | No | 5.1415 | 1.5045 | - | - |
> > > | 8  | Yes | 5.1589 | 1.4990 | 82.16 | 13.67 |
> > > | 32 | No | 6.2479 | 4.5370 | - | - |
> > > | 32 | Yes | 6.2923 | 4.5602 | 161.40 | 17.12 |
> > >
> > > [1] DeepSeekMath: Pushing the Limits of Mathematical Reasoning in Open Language Models, arXiv 2024

---

> ### Comment · Reviewer_WR7w · 2025-08-08
>
> Thank you for the authors’ rebuttal. Your response has addressed most of my concerns.
> 1. For Q1, should it correspond to a drop in street-level accuracy rather than continent-level accuracy?
> 2. For Q2, I have fully understood your explanation.
> 3. For Q3, I was expecting a comparison of GLOBE’s efficiency with other baseline methods. Even if GLOBE is less efficient than others (which is understandable), I still hope the authors can include this part in the final version for readers’ reference.
>
> I will update my rating accordingly.

---

> > ### Author Response · Authors · 2025-08-08
> >
> > Thank you for the follow-up and rating update.
> >
> > For **Q1**, we would like to clarify why continent-level accuracy can also be affected by coordinate query errors. When the predicted location name is correct at the city or country level, the geocoding API occasionally returns a mismatched coordinate due to name ambiguities. For example, if the ground truth is Paris, France (48.8566° N, 2.3522° E) but the API instead returns Paris, Texas, USA (33.6609° N, −95.5555° W), the Haversine distance between them exceeds the continent-level threshold of 2,500 km. Although such cases are rare, they can substantially lower continent-level accuracy because the evaluation is based on these queried coordinates rather than directly predicted GPS points, making both street-level and continent-level metrics more sensitive to such mismatches.
> >
> > For **Q3**, we appreciate the suggestion and will include a runtime comparison with other baselines in the final version. The table below provides an illustrative example of such a comparison, reporting latency, throughput, and streaming metrics, with all models tested under the same hardware (single H20 GPU) and software settings (e.g., PyTorch, vLLM) as GLOBE.
> >
> > ### Table 6. Inference efficiency comparison between different baseline methods
> >
> > | Model                 | Concurrency (batch size) | Streaming | Avg. Latency (s) | Throughput (QPS) | TTFT (ms) | TPOT (ms) |
> > | --------------------- | ------------------------ | --------- | ---------------- | ---------------- | --------- | --------- |
> > | GeoCLIP               | 1                     | –         | 0.1364           | 7.3313           | –         | –         |
> > | PLONK                 | 1                        | –         | 0.5852           | 1.7088           | –         | –         |
> > | Qwen2.5-VL-7B    | 1               | No        | 2.9368           | 0.3405           | –         | –         |
> > | InternVL3-8B    | 1                     |No         | 3.4886           | 0.2866           | –         | –         |
> > | GLOBE (InternVL3-8B)  | 1         | No        | 4.8742           | 0.2051           | –         | –         |
> > | GLOBE (InternVL3-8B ) | 1        | Yes       | 4.8597           | 0.2057           | 64.23     | 11.70   |
> > | GLOBE (Qwen2.5-VL-7B) | 1    | No        | 4.5684           | 0.2188           | –         | –         |
> > | GLOBE (Qwen2.5-VL-7B) | 1    | Yes       | 4.5628           | 0.2191           | 34.67     | 11.72     |

---

### Official Review · Reviewer_T7Q7 · 2025-06-27

**Clarity:** 2
**Significance:** 2
**Originality:** 3
**Rating:** 5
**Confidence:** 4

**Summary:**

This paper introduces a benchmark and a reinforcement learning-based method for image geolocation. It proposes a curation process to select training samples that are already well localized by a particular vision-language model and incorporates reward functions to encourage semantic consistency and precise localization. The geolocalization process is cast as a reinforcement learning problem, with reward definitions that are claimed to align with the task objectives. They evaluate on their dataset, as well as an external and small-scale geolocation dataset.

**Questions:**

**Questions:**
- Do you agree with the criticism of the circular logic of using the same model to curate and evaluate?
- What is the size of the MP16 reason train/test? How diverse is it geoographically?
- How does the rewards Rloc encourage anything meaningful when it only promotes predicting that images are localizable?
- What is the rationale of using RL instead of just fine-tuning / LORA-ing the models? What si the rationale of using GRPO?
- various missing details, see weaknesses

**Remarks:**
- The acronym GLOBE ("... Bi-objective geo-Enhancement") feels forced.
- The paper opens with a reference to an 18-year-old work to motivate recent interest, which weakens the relevance argument.
- All figures should be provided as vector graphics, not raster images (PNGs).
- The authors use "locatability" instead of the accepted term "localizability," as coined in [3].
- Citation for GeoCLIP's confidence score appears to be mismatched or incorrectly formatted ("that1 quantifies locatability [8]"), and should likely be [46].
- The distinction between geolocation and recognition is not clearly addressed, these are not interchangeable tasks.

**Ethical Concerns:**

["NO or VERY MINOR ethics concerns only"]

**Final Justification:**

All concerns have been addressed, including the major one about circularity. The paper, with the new experiments, appears ready for publication.

**Limitations:**

Well addressed

**Quality:**

3

**Strengths And Weaknesses:**

**Strength:**

- The task and proposition of the paper (evaluating Chain of thoughts for geolocation) are very interesting
- Reinforcement learning for geolocation is a novel and underexplored angle.
- It introduces a curated benchmark, which, if clearly defined and reproducible, could be valuable for the community.
- Extensive ablation study

**Weaknesses:**
- The curation process selects images that are already well-localized by the very models used as backbones during evaluation (QWEN), creating a potential feedback loop and an unfair advantage. This setup risks confirming performance gains that are a direct consequence of the initial filtering. Furthermore, the filtering method propagates the pre-existing biases of the LLMs (e.g., geographic coverage biases).

- Even when selecting images based on performance from Qwen, and using Qwen as a backbone, the model is still outperformed by ChatGPT, which is not clearly explained.

- The reward function R_loc only encourages predicting that images *are* localizable. There is no mechanism to predict or reward correctly identifying non-localizable images, especially since all training images are selected to be localizable, effectively training the model to always say "localizable." So this reward may be meaningless?

- The authors do not track performance on intermediate tasks such as localizability prediction or semantic consistency of the reasoning, despite emphasizing these as key contributions.

- Lack of crucial information, such as the size and geographical distribution of the proposed dataset.

- Several key methodological steps are glossed over or vaguely described, including:
  - The phrase "semantically aligned reasoning chains are retained" is never clarified.
  - The method for measuring alignment between reasoning trajectories and visual segmentation is not explained.
  - The multimodal aspect of the "robust reasoning" is unclear.
  - The Match and DKL functions are used without formal definitions.
  - The notion of "groups" is never formally explained, and the nature of the "prompt groups" is unclear

- The claim that all previous geolocation methods are either classification or retrieval ignores existing regression-based [1] or generative approaches [2,3], which are also not included in the experimental comparison.

- The rationale for using reinforcement learning over simpler fine-tuning methods is quite vague: "guide complex reasoning behaviors in a more structured and interpretable manner". How does this structure and interpretability translate for your task? This is poorly motivated and not evaluated.  Similarly, the usefulness of group-based rewards is not justified or evaluated (using PPO instead og GRPO).

- The authors do not report distance-based metrics (eg, average Haversine) to evaluate the precision of their model, as is usually done for comparable studies

- The potential influence of chain-of-thought (CoT) reasoning, which is central to the proposed method, is not evaluated, as all evaluated models use it.

[1] OpenStreetView-5M: The Many Roads to Global Visual Geolocation, CVPR2024
[2] Around the World in 80 Timesteps: A Generative Approach to Global Visual Geolocation, CVPR2025
[3] Exploiting the Earth's spherical geometry to geolocate images, ECML2019

**Overall**:
The paper targets an important problem and introduces some potentially valuable ideas. However, serious methodological weaknesses, a lack of justification/evaluation of design decisions, and unclear evaluation choices significantly reduce its impact. The proposed benchmark suffers from circular logic due to using the same model used to curate the dataset as the backbone. Key components are insufficiently defined (eg, the size of the proposed dataset!), and the reinforcement learning formulation lacks proper justification. The work would benefit from clearer exposition, more rigorous evaluation, and a thorough comparison with existing non-classification, non-retrieval, non LLM, non CoT methods. At this stage, the contribution does not convincingly support its claims.

---

> ### Author Rebuttal · Authors · 2025-07-30
>
> Thank you for your detailed review and critical feedback. We appreciate your positive assessment of our $\textbf{novel idea}$, $\textbf{curated benchmark}$, and $\textbf{extensive ablation study}$. We respect your concerns and provide clarifications below to address the key misunderstandings and better convey the contributions of our work.
>
> > ### 1. The explanation of data curation and the biases of the LLMs (W1 & Q1)
>
> A1: We clarify that the models used for data curation are $\textbf{distinct}$ from the training backbone, both in scale and capability. Specifically, our distillation sources include $\textbf{Qwen2.5-VL-72B}$ and InternVL3-78B, while the student model is $\textbf{Qwen2.5-VL-7B}$, a significantly smaller and less capable architecture, as documented in the Qwen2.5-VL technical report [1].
>
> We emphasize that our data curation is a controlled distillation process aimed at extracting high-quality, reasoning-rich samples, not a source of bias. Rather than promoting memorization, it is designed to activate the model’s reasoning ability and leverage internal world knowledge, even with limited visual diversity. For a detailed explanation, please refer to our response to Reviewer wDfp in A2.
>
> > ### 2. The discussion of GLOBE and GPT4.1 (W2)
>
> A2: While our method performs strongly among open-source LVLMs, we acknowledge that it does not surpass closed-source models like GPT-4.1, an expected gap due to differences in scale and resources. Our aim is to advance open, reproducible, and data-efficient LVLM training. Notably, GLOBE (7B) outperforms the much larger Qwen2.5-VL-72B on both MP16-Reason-Test and IMG2GPS3K, demonstrating the effectiveness of our distillation framework. For details, please refer to our response to Reviewer wDfp in A3.
>
> > ### 3. The explanation of the reward function $R_{loc}$ (W3 & Q3)
>
> A3: We apologize if our original description of the $R_{loc}$ reward function was unclear and may have led to a misunderstanding. We appreciate the opportunity to clarify its design and importance.
> R_loc is not intended to trivially encourage all outputs to be "localizable." Although the training set is curated to contain generally localizable images, the key idea behind R_loc is to assess the locatability of an image–reasoning pair, rather than the image alone. During GRPO training, multiple reasoning traces are sampled for each image, and $R_{loc}$ is computed based on both the image and its generated explanation. As a result, different reasoning traces lead to different $R_{loc}$ scores.
> This mechanism implicitly encourages the model to generate higher-quality, more location-grounded reasoning.
>
> > ### 4. The tracking performance on intermediate tasks (W4)
>
> A4: As figures cannot be included in the rebuttal process, we present representative data in Table 1.
> ### Table 1. Performance on intermediate tasks
> | Reward | Step 1 | Step 200 | Step 500 | Step 1000 | Step 1500 |
> | ------- | -------- | -------- | --------- | -------- | -------- |
> | Locatability  | 0.72 | 0.76 | 0.81 | 0.84 | 0.84 |
> | Visual Grounding Consistency  | 0.35 | 0.51 | 0.63 | 0.66 | 0.67 |
>
> The evaluation rewards exhibit steady improvement throughout training, with optimal performance observed at Step 1500.
> We observe that both rewards increase steadily during training, showing consistent improvement and stabilizing by Step 1000, with only marginal gains thereafter. By Step 1500, both reach their peak values, indicating convergence.
> We will also update charts showing the reward score trajectories throughout training to illustrate the optimization dynamics better.
>
> > ### 5. The clarification of the proposed dataset (W5 & Q2)
>
> A5: We agree that dataset statistics and geographical distribution are important for understanding the properties of MP16-Reason. As shown in Table 2, we have already reported key statistics of the dataset, including the number of images in the training and test sets, as well as the distribution across cities and countries. To further improve transparency, in the revised manuscript, we will include a geographical heatmap to illustrate the spatial coverage of both the training and test splits.
>
> ### Table 2. Statistics of the proposed dataset
> | Data | #Samples | #Countries | #Cities | #Indoor scene | #Natural scene | #Urban scene |
> | ------- | -------- | -------- | --------- | -------- | -------- | -------- |
> | MP16-Reason-Train  | 33721 | 134 | 1944 | 5393 | 2077 | 26251 |
> | MP16-Reason-Test | 12000 | 145 | 3012 | 2096 | 1092 | 8812 |
>
> '#' indicates number of.
>
> > ### 6. The clarification of the methodology section (W6)
>
> A6: We thank the reviewer for the thoughtful feedback. Several methodological components, such as reasoning alignment, entity matching,  and group definition, are central to our approach. To enhance clarity and accessibility for a broader audience, we will revise the Methodology section in the final version to include more precise definitions, improved descriptions, and illustrative examples.
>
>
> > ### 7. The supplementary comparison experiments (W7)
>
> A7: We thank the reviewer for pointing out the additional related work [2, 3, 4]. We will update both the Related Work section and the main results table accordingly (Since [4] does not have an open-source model, we only compared methods [2] and [3] on MP16-reasoning-Test and IMG2GPS3k in Table 3 and 4). The experimental results also show that the performance of the model trained only with street view data will be poorer in diverse scenarios, which is consistent with the viewpoint proposed in our work.
>
> ### Table 3. Performance comparison on MP16-reasoning-Test
> | Method | Street (1km) | City (25km) | Region (200km) | Country (750km) | Continent (2500km) |
> | ------- | -------- | -------- | --------- | -------- | -------- |
> | PLONK (iNaturalis) [3] | 0.00 | 0.75 | 4.11 | 17.06 | 40.6 |
> | PLONK (OSV_5M) [3] | 0.08 | 15.95 | 41.75 | 69.87 | 87.29 |
> | PLONK (YFCC) [3] | 11.72 | 46.64 | 60.46 | 77.97 | 91.96|
> | RFM (OSV_5M) [2] | 0.97 | 16.53 | 28.72 | 50.31 | 71.47|
> | GLOBE (ours) | 17.38 | 58.06 | 70.91 | 83.82 | 90.75 |
>
>
> ### Table 4. Performance comparison on IMG2GPS3k
> | Method | Street (1km) | City (25km) | Region (200km) | Country (750km) | Continent (2500km) |
> | ------- | -------- | -------- | --------- | -------- | -------- |
> | PLONK (iNaturalis) [3] | 0.00 | 1.03 | 4.70 | 15.68 | 39.81|
> | PLONK (OSV_5M) [3] | 0.07 | 9.21 | 28.16 | 50.48 | 74.21|
> | PLONK (YFCC) [3] |5.41 | 29.70 | 44.71 | 61.83 | 79.55|
> | RFM (OSV_5M) [2] | 0.83 | 13.28 | 25.33 | 43.84 | 65.63 |
> | GLOBE (ours) | 9.84 | 40.18 | 56.19 | 71.45 | 82.38|
>
>
>
> > ### 8. The reason for using GRPO-based RL ( W8 &  Q4)
>
> A8: We clarify that our choice of RL, specifically Group-based Reward PPO (GRPO), over standard supervised fine-tuning (SFT) is motivated by the need to guide multi-step, structured reasoning rather than simply mimic surface-level patterns in the training data.
>
> In SFT, the model is trained to reproduce reference outputs (e.g., reasoning traces), which often leads to shallow imitation and limited generalization, especially when the test inputs differ in structure or context.  In contrast, RL with carefully designed rewards allows us to shape intermediate behaviors: for example, encouraging the model to first identify geographic clues (e.g., vegetation, architecture), then narrow down the region. This stepwise reasoning is reflected in the generated traces and makes the decision process more interpretable.
> GRPO, in particular, improves upon standard PPO by optimizing based on relative preferences within a group of sampled responses, rather than absolute reward signals. As shown in prior work [5], this leads to more stable training and better alignment with human (or model-based) judgment, especially when rewards are sparse or noisy, as in reasoning tasks.
>
> > ### 9. The distance-based metrics for evaluation (W9)
>
> A9: We clarify that the benchmark Img2GPS3K is evaluated using distance-based metrics. Since GLOBE predicts semantic location labels (e.g., city and country), we report accuracy as the primary metric. However, we can also provide distance-based results (See Table 3 and 4) using an external geocoding tool (Microsoft Azure Maps tool).
>
> >  ### 10. Additional ablation experiments for CoT (W10)
>
> A10: We conducted ablation studies on two backbones (Qwen2.5-VL-7B, InternVL3-8B) comparing with and without CoT. Table 5 shows consistent but modest gains, indicating the positive influence of CoT. We will include this analysis in the revision.
>
> ### Table 5. Performance comparison on MP16-reasoning-Test
> | Backbone | CoT | Street (1km) | City (25km) | Region (200km) | Country (750km) | Continent (2500km) |
> | ------- | -------- | -------- | --------- | -------- | -------- | -------- |
> | Qwen2.5-VL-7B| ✗ | 14.37 | 51.11 | 61.29 | 73.67 | 82.46|
> | Qwen2.5-VL-7B| ✓ | 15.42 | 52.72 | 62.86 | 75.11 | 83.47|
> | InternVL3-8B | ✗ | 11.45 | 42.66 | 54.64 | 75.17 | 86.67|
> | InternVL3-8B | ✓ | 12.01 | 44.17 | 55.66 | 75.36 | 86.98|
>
> > ### 11. Some remarks (Remarks)
>
> A15: We thank the reviewer for these helpful remarks. We will carefully address each point in the revision, including acronym refinement, citation corrections, terminology consistency, figure format improvements, and clarification of task definitions.
>
>
> [1] Qwen2.5-VL Technical Report, arXiv 2025
> [2] OpenStreetView-5M: The Many Roads to Global Visual Geolocation, CVPR2024
> [3] Around the World in 80 Timesteps: A Generative Approach to Global Visual Geolocation, CVPR2025
> [4] Exploiting the Earth's spherical geometry to geolocate images, ECML2019
> [5] Deepseek-r1: Incentivizing reasoning capability in LLMs via reinforcement learning, arXiv 2025

---

> > ### Comment · Reviewer_T7Q7 · 2025-08-04
> > **Response to rebuttal**
> >
> > The reviewer thanks the authors for their rebuttal, which reflects a tremendous amount of work and careful consideration. Below is a detailed account of the reviewer’s opinion regarding the authors’ responses:
> >
> > - The reviewer had not initially realized that two different QWEN models were used: one for generating ground truth and another as the predictive backbone. This only partially addresses concerns about circularity, as the reviewer remains concerned given that both models come from the same family and were distilled fom the same model and dataset.Ttherefore, the risk of circularity, while somewhat mitigated, still stands. It would be important to address this with a backbone swap. If the reviewer's understanding is correct, this was done with Intern?
> >
> > - The reviewer fully understands that a public model may not compete with a proprietary one trained on unknown data sources. This is entirely acceptable, but the authors should make this point as explicit in the paper as they have in the rebuttal.
> >
> > - Regarding the additional experiments: these are highly appreciated and provide valuable insights. However, some details remain unclear:
> >
> >     - PLONK: Is this referring to the Riemannian Flow Matching approach of [3]?
> >
> >     - RFM (OSV_5M) [2]: Which specific model from [2] is being evaluated here? Is it the hybrid regression model? The term “Riemannian Flow Matching (RFM)” does not appear in [2], so the reviewer is unsure what is meant by RFM [2].
> >
> > - On the distance metric: the reviewer had in mind a more standard geolocation metric (reporting the average Haversine distance between true and predicted coordinates) rather than a quantized percentage-based approach. This would better align with established geolocation evaluation practices.
> >
> > The reviewer appreciates the authors’ thoughtful answers and is inclined to improve their score. However, before doing so, the reviewer would like to confirm that the nomenclature used for the competing models in Tables 3 and 4 is correct, as there is a strong suspicion that an error may have occurred in the labeling of methods.

---

> > > ### Author Response · Authors · 2025-08-05
> > >
> > > We sincerely thank the reviewer for your thoughtful and constructive feedback. We are grateful for your recognition of our efforts and appreciate your willingness to increase the score. Below, we provide clarifications to your remaining concerns.
> > >
> > > > ### Q1: The discussion of circularity
> > >
> > > We understand the reviewer’s caution in using models from the same family for distillation and student training. However, we emphasize that our distillation pipeline targets generalizable reasoning patterns, not model-specific behaviors.
> > >
> > > To demonstrate that our method enables cross-model knowledge transfer, rather than relying on architectural alignment, we refer to Table 6, originally provided in our response to Reviewer WR7w. It summarizes ablation results using different data sources for training the Qwen2.5-VL-7B student model.
> > >
> > > Notably, even when the training data is distilled solely from InternVL3-78B, a model with a different architecture and pretraining lineage, GRPO (with Qwen2.5-VL-7B backbone) still yields significant improvements over the baseline, whereas SFT does not show clear gains. This highlights the effectiveness of our proposed GRPO training strategy compared to standard supervised fine-tuning.
> > >
> > > Moreover, the best performance is achieved with our multi-VLM distilled dataset (MP16-Reason-Train), indicating that gains stem from high-quality, diverse supervision, not circular influence from shared model ancestry. We will include this analysis in the revised manuscript to enhance clarity.
> > >
> > > ### Table 6. Performance comparison using different Training datasets on MP16-Reason-Test, with Qwen2.5-VL-7B
> > >
> > > | Training Dataset | Training Strategy | Street (1km) | City (25km) | Region (200km) | Country (750km) | Continent (2500km) |
> > > | -- | -- | -- | -- | -- | -- | -- |
> > > | Baseline | - | 15.42 | 52.72 | 62.86 | 75.11 | 83.47 |
> > > | Random | SFT | 15.23 | 52.00 | 64.56 | 78.17 | 85.23 |
> > > | Random | GRPO |17.26 | 59.22 | 71.80 | 84.73 | 91.26 |
> > > | Only InternVL3-78B distill | SFT | 15.22 | 52.47 | 65.09 | 78.79 | 86.15 |
> > > | Only InternVL3-78B distill | GRPO |17.37 | 59.45 | 71.88 | 84.74 | 91.24 |
> > > | MP16-Reason-Train | SFT | 16.38 | 56.76 | 70.21 | 83.82 | 90.75 |
> > > | MP16-Reason-Train | GRPO |17.99 | 62.85 | 73.83 | 86.68 | 92.52 |
> > >
> > > > ### Q2: The clarification of GLOBE compared to closed-source models
> > >
> > > Thank you for this suggestion. We fully agree and will add a clear statement in the Experiment section of the paper, this will ensure readers have appropriate expectations and better understand the context of our comparisons.
> > >
> > > > ### Q3: The clarification of comparison models
> > >
> > > Thank you for the clarification.
> > > Yes, PLONK refers to the Riemannian Flow Matching method of [3], and we follow the model names from the official HuggingFace repository (PLONK trained on iNaturalist, OSV-5M, and YFCC).
> > > For [2], we evaluated the hybrid regression model publicly released from the HuggingFace repository (**osv5m/baseline**), not RFM; we apologize for the incorrect labeling and will correct it to "Hybrid Regression (OSV-5M) [2]" later.
> > >
> > > > ### Q4:  The discussion of metric
> > >
> > > We fully agree that the average Haversine distance is a standard and informative metric in geolocation evaluation.
> > > In response, all results presented in our rebuttal use this distance-based metric, which allows for a more fine-grained and comparable assessment across methods. We will update all tables in the manuscript to adopt this metric in the final version.

---

> > > > ### Comment · Reviewer_T7Q7 · 2025-08-06
> > > > **Follow up**
> > > >
> > > > Thank you for your quick answer. The reveiwer's concerns have all been answer, and the score will be changed accordingly.

---

> > > > > ### Author Response · Authors · 2025-08-06
> > > > >
> > > > > Thank you for your helpful suggestions during the discussion period.

---

### Official Review · Reviewer_Napy · 2025-07-03

**Clarity:** 2
**Significance:** 3
**Originality:** 3
**Rating:** 4
**Confidence:** 3

**Summary:**

This paper introduces GLOBE, a reasoning-driven image geo-localization method using Large Vision-Language Models (LVLMs). This work addresses dataset diversity and reasoning limitations by curating MP16-Reason, a social media-based dataset with distilled reasoning traces. The authors then propose GLOBE, an LVLM fine-tuned using Group Relative Policy Optimization (GRPO), which is a reinforcement learning framework that optimizes three task-specific rewards: locatability reward, visual grounding consistency reward, and geo-localization accuracy reward. The proposed method outperforms state-of-the-art models in accuracy while being data-efficient (33K samples).

**Questions:**

Please see weaknesses. Overall, I think this is a good paper. Since I am not an expert in this area, I am currently giving a "Borderline Accept" rating and am willing to update the score after the Rebuttal.

**Ethical Concerns:**

["NO or VERY MINOR ethics concerns only"]

**Final Justification:**

The author has addressed most of my concerns. I also read the comments of other reviewers. Finally, I decide to maintain my original positive score. However, it is very necessary that the authors update the final version based on the reviewers' comments and the author's own rebuttal, including the explanation of fine-grained localization, more discussion on related work, and the clarification of computational costs/requirements in both training and inference.

**Limitations:**

Limitations are discussed in Appendix A.2.

**Quality:**

3

**Strengths And Weaknesses:**

**Strengths**

1. Novel Dataset (MP16-Reason). This work addresses the lack of reasoning-oriented geo-localization datasets by curating diverse social media images with model-derived reasoning traces. It incorporates multi-model distillation and verification to ensure high-quality annotations.

2. Innovative Methodology (GLOBE). This work introduces GRPO-based fine-tuning, which optimizes multiple reasoning-specific rewards (locatability, visual grounding, and geo-localization). It demonstrates that structured rewards improve interpretability and grounding in model reasoning.

3. Reasonable Motivation. The analysis of reasoning-oriented geo-localization using LVLM (such as the limitations of previous methods) is clear and reasonable.

3. Strong Results. The experiments show that the proposed method outperforms existing LVLMs (Except for GPT-4.1) and retrieval/classification baselines on benchmarks like MP16-Reason-Test and IM2GPS3K. It shows robustness across diverse scenes.

4. Data Efficiency. The work achieves strong performance with only 33K training samples, suggesting that targeted reasoning supervision is more effective than large-scale but generic datasets.

**Weaknesses**

1. Fine-Grained Localization Limitations. The model struggles with coordinate-level geo-localization, as reasoning relies on high-level semantic cues (e.g., architecture, language) that lack spatial specificity. The authors acknowledged this, but I think it's not enough. Firstly, other geo-localization works have reported accuracy at higher localization precision, at least street-level (1km). This paper lacks results at this precision. Secondly, retrieval-based methods can achieve better accuracy at higher precision (several meters), and the authors claim that they are considering combining reasoning with retrieval to improve fine-grained accuracy in future work. I think that in the second paragraph of the Introduction section, it should be clearly stated that the retrieval method has this advantage.

2. In the Related Work section, the introduction to retrieval-based geo-localization is all cross-view geo-localization. However, the more general visual geo-localization (aka Visual Place Recognition) was not mentioned. I think at least the following classic or SOTA methods should be mentioned:
- NetVLAD: CNN architecture for weakly supervised place recognition. CVPR 2016.
- Rethinking visual geo-localization for large-scale applications. CVPR 2022.
- CricaVPR: Cross-image correlation-aware representation learning for visual place recognition. CVPR 2024.
- BoQ: A place is worth a bag of learnable queries. CVPR 2024.

3. Computational Costs. The paper does not quantify the resources required for multi-model distillation or GRPO fine-tuning, which is not conducive to other researchers following up on this work.

4. Dependence on Distilled Knowledge. MP16-Reason relies on reasoning traces distilled from other LVLMs (e.g., Qwen2.5-VL, InternVL3), which may inherit biases or errors from these models. The verification process mitigates this but may not eliminate the risk entirely.

5. Limited Performance Against GPT-4.1. While GLOBE outperforms open-source LVLMs, comparisons to proprietary models like GPT-4.1 are limited (Table 1 shows GPT-4.1 performs better). I think this is acceptable, but it is suggested that the author provide more discussion on this matter.

---

> ### Author Rebuttal · Authors · 2025-07-30
>
> Thank you for the positive assessment and constructive feedback. We appreciate your support for our contributions ($\textbf{Novel Dataset}$, $\textbf{Innovative Methodology}$, $\textbf{Reasonable Motivation}$, $\textbf{Strong Results}$, and $\textbf{Data Efficiency}$) and address below the remaining concerns to further clarify and strengthen our work.
>
> > ### 1. The explanation of fine-grained localization (W1)
>
> A1: We thank the reviewer for this insightful comment. We fully agree that fine-grained localization is a critical capability, and we appreciate the opportunity to clarify and strengthen the evaluation of our work.
> Regarding localization precision, in this rebuttal process, all reported results are now computed using a high-precision evaluation pipeline. Specifically, we use Microsoft Azure Maps to convert predicted city names into precise GPS coordinates and compute the Haversine distance between the predicted and ground truth locations. Accuracy is then evaluated at five standard levels: Street (1km), City (25km), Region (200km), Country (750km), and Continent (2500km), following established protocols in prior work.
> Although performance at the 1km level is inherently limited (since our model operates at the city level or coarser), our approach achieves competitive results compared to other open-source methods at the city level and above. This demonstrates the effectiveness of our reasoning-driven framework within its intended scope (see our response to Reviewer T7Q7 and the updated results in Table 3 and Table 4).
> Second, we agree that retrieval-based methods can achieve significantly higher precision and are particularly effective for fine-grained localization. As suggested, we will revise the second paragraph of the Introduction to explicitly acknowledge this strength of retrieval approaches, and better position our method as complementary
>
> > ### 2. The supplementary related work (W2)
>
> A2: We thank the reviewer for pointing this out. We will revise the Related Work section to incorporate the suggested works and improve its overall completeness.
>
> > ### 3. The clarification of computational costs (W3)
>
> A3: For data curation, we deployed Qwen2.5-VL-72B and InternVL3-78B using 8×H20 GPUs under the VLLM framework,  while GeoCLIP was run separately on a single H20 GPU. All models performed inference over the original MP16 dataset.
> For GRPO training, each experiment with our 7B-sized model was conducted using 8×H20 GPUs. With a batch size of 16, the training throughput was approximately 0.44 examples per second.
>
> > ### 4. The explanation of bias in MP16-Reason (W4)
>
> A4: We emphasize that our data curation is not a source of bias, but a controlled knowledge distillation process designed to extract high-quality, informative samples for efficient LVLM training. This approach aligns with recent advances in the literature, where large models are used to construct compact, high-signal training sets for downstream tasks [1, 2, 3].
> Crucially, the goal of distillation is not to replicate the full data distribution, but to identify a representative and semantically rich subset that maximizes learning efficiency. By prioritizing quality over quantity, our method enables state-of-the-art performance with significantly reduced training data and cost.
> $\textbf{Moreover, our objective is not to encourage memorization of specific samples}$, but to activate the model’s reasoning capabilities through well-structured, high-signal examples. These curated instances serve as cognitive anchors that guide the model to internalize spatial and contextual reasoning patterns, thereby promoting generalization, leveraging the model’s internal world knowledge even in the presence of limited visual diversity.
>
> > ### 5. The discussion of GLOBE and GPT-4.1 (W5)
>
> A5: While our method achieves strong performance among open-source LVLMs, we acknowledge that it does not surpass closed-source models such as GPT-4.1. This gap is not unexpected, as industrial-scale systems benefit from vastly larger training datasets, proprietary infrastructure, and orders-of-magnitude more compute resources, typically beyond the scope of academic research.
> Our goal is not to compete directly with such closed systems, but to advance open, reproducible, and data-efficient LVLM training. Within this context, our approach demonstrates significant progress. Notably, GLOBE, built upon the Qwen2.5-VL-7B backbone, outperforms the much larger Qwen2.5-VL-72B (the very model used to generate the distilled data) on the MP16-Reason-Test and IMG2GPS3k benchmark.
> This "student surpassing teacher" result highlights the effectiveness of our distillation and training framework in extracting and refining high-quality knowledge, rather than merely replicating teacher behavior. It underscores the value of curated, reasoning-grounded supervision in boosting model capabilities beyond scale.
> To promote transparency and community advancement, we will release all $\textbf{code}$, $\textbf{models}$, and $\textbf{curated data}$.
>
>
> [1] LESS: selecting influential data for targeted instruction tuning. ICML 2024
> [2] LIMA: Less Is More for Alignment, NeurIPS 2023
> [3] Jiuzhang3.0: Efficiently improving mathematical reasoning by training small data synthesis models, NeurIPS 2024

---

> ### Comment · Reviewer_Napy · 2025-08-04
>
> Thank you for the detailed responses to my concerns. I have no further questions. I strongly encourage that the authors update the final version based on the reviewers' comments and the author's own rebuttal, including the explanation of fine-grained localization, more discussion on related work, etc.

---

> > ### Author Response · Authors · 2025-08-05
> >
> > Many thanks again for your thoughtful feedback throughout the review process.

---

### Official Review · Reviewer_WWdG · 2025-07-03

**Clarity:** 3
**Significance:** 3
**Originality:** 3
**Rating:** 4
**Confidence:** 4

**Summary:**

This paper proposes a new pipeline for reasoning-driven image geo-localization leveraging LVLMs. The key components include the creation of MP16-Reason, a curated dataset containing diverse social media images with explicit reasoning supervision, and GLOBE, a fine-tuned LVLM using Group Relative Policy Optimization (GRPO) for improved multimodal reasoning and geo-localization. GLOBE incorporates multiple reward signals to balance locatability, visual grounding, and localization accuracy. Empirical results across multiple datasets show notable improvements over prior methods, both in quantitative accuracy and the interpretability of reasoning trajectories.

**Questions:**

1. Can the authors provide quantitative data or examples on the diversity/independence of reasoning chains in MP16-Reason, especially in comparison to single-model annotation or using only existing datasets?

2. Could the authors elaborate on failure cases, especially those involving ambiguous or visually similar locations? Are there examples where reasoning appears correct but leads to systematic error, or where the distilled supervision might mislead?

3. Is the reward weight selection (λ1, λ2, λ3) in the composite reward function fixed or tuned, and how sensitive is the training pipeline to these relative weights? Any guidance for future extensions?

4. How would the model perform on zero-shot generalization to out-of-domain images or adversarially perturbed content? Has any robustness testing been conducted?

**Ethical Concerns:**

["NO or VERY MINOR ethics concerns only"]

**Final Justification:**

Thanks for the rebuttal with additional analysis. The results have addressed my concerns, and I have updated the score.

**Limitations:**

Yes

**Paper Formatting Concerns:**

No.

**Quality:**

3

**Strengths And Weaknesses:**

Strengths:

1. The MP16-Reason dataset advances the field by providing richly annotated, diverse social media imagery with explicit, model-distilled reasoning chains.

2. The paper introduces a structured reward design and GRPO-based reinforcement learning tailored for geo-localization with LVLMs. The composite reward system targets known limitations in SFT approaches and links reasoning paths to prediction reliability.

3. The model produces step-by-step reasoning traces grounded in visual evidence. Figure 2 and Figure 5 provide illustrative qualitative examples, highlighting the model’s interpretability and ability to justify location decisions. This is especially valuable for both debugging and trustworthiness in practical scenarios.

4. Across both the proprietary MP16-Reason-Test set and the public IM2GPS3K benchmark, GLOBE demonstrates improved performance to prominent open-source models.

Weaknesses

1. While the multi-model distillation and verification pipeline is rigorous, supervision is derived from existing LVLMs that themselves may not surpass or generalize beyond their own knowledge boundaries and biases. There remains a risk that reasoning in MP16-Reason could amplify inherited model biases, or fail on truly out-of-domain or rare cases not seen by the distilling models. It is not fully assessed how diverse or independent the distilled reasoning is, nor is its possible narrowness quantified.

2. Although the authors claim superior scene diversity, there is little detailed breakdown of performance across different image types or conditions, which limits the ability to judge the external validity of the approach.

3. Section A.1.3 provides basic settings, but the selection of key parameters for the GRPO framework, particularly the reward weights (λ parameters), lacks detailed justification or sensitivity analysis.

4. Only one proprietary model (GPT-4.1) is included as the baseline and there is still a performance gap between *GLOBE* and GPT-4.1. Maybe introducing more potential baselines in such types could demonstrate the performance of current SOTA methods. Furthermore, the paper does not explicitly show the parameter size of *GLOBE*.

5. Section A.3 addresses privacy concerns, but the paper could benefit from a more substantial analysis of misuse risks, such as adversarial image manipulations, regional demographic impacts, or surveillance scenarios. No empirical robustness experiments are presented.

---

> ### Author Rebuttal · Authors · 2025-07-30
>
> Thank you for the time and effort in reviewing our work. We sincerely appreciate your constructive feedback, which has helped us identify areas for improvement. We are also grateful for your recognition of our **curated dataset**, **the novelty of our approach**, and the **strong performance** demonstrated in the experiments. Below, we address the concerns raised and clarify key aspects of our contributions.
>
> > ### 1. The explanation of bias in MP16-Reason (W1)
>
> A1: We emphasize that our data curation is not a source of bias, but a controlled knowledge distillation process designed to extract high-quality, informative samples for efficient LVLM training. This approach aligns with recent advances in the literature, where large models are used to construct compact, high-signal training sets for downstream tasks [1, 2, 3].
> The goal of distillation is not to replicate the full data distribution, but to extract a representative, high-signal subset that enhances learning efficiency. Rather than promoting memorization, our curated examples are designed to activate reasoning and guide the model to generalize via internal world knowledge, even with limited visual diversity.
> For a detailed explanation, please refer to our response to Reviewer wDfp in A2.
>
> > ### 2. The clarification of performance across different conditions (W2)
>
> A2: To better evaluate the generalization and external validity of our approach across diverse visual scenes, we have conducted a fine-grained breakdown of performance on the MP16-Reason-Test. Following the MP-16 dataset [4], we categorize images by scene condition (e.g., indoor, natural, urban) based on metadata that includes predicted scene probabilities. For each image, we assign the label with the highest predicted probability as its scene classification.
>
> ### Table 1. Performance comparison across different scene conditions on MP16-Reason-Test. Each entry reports distance-based localization accuracy (km) at multiple scales: Street (1km) / City (25km) / Region (200km) / Country (750km) / Continent (2500km).
>
> | Model / Scene | Indoor | Nature | Urban |
> | --- | --- | --- | --- |
> | Qwen2.5-VL-7B | 12.50/46.95/55.30/69.99/81.49 | 8.61/42.77/60.07/72.62/80.68 | 16.95/55.32/65.00/76.63/84.29 |
> | GeoReasoner | 12.57/35.93/48.50/65.87/79.04 | 5.10/35.71/48.98/67.35/78.57 | 10.18/42.23/51.86/68.78/80.19 |
> | GLOBE (Ours) | 17.65/57.35/64.71/80.88/91.18 | 13.95/55.81/81.40/90.70/97.67 | 18.61/64.98/74.76/87.38/92.11 |
>
> It can be found that GeoReasoner performs best in urban scenes, which is closely related to its training on the Street View dataset. Among them, our method consistently achieves the highest performance across all scene types, particularly excelling in nature scenes with substantial gains over both baselines. This detailed analysis confirms our claim of superior scene diversity and supports the broader applicability of our approach.
>
> > ### 3. The clarification of key parameters (λ) in GRPO (W3 & Q3)
>
> A3: The weights  λ₁, λ₂, and λ₃ are fixed during training and were initially set to 0.2, 0.5, and 1, respectively. Since the primary objective is to accurately predict the city and country, we assigned the highest weight to λ₃, which directly supervises geolocation accuracy. Consistency is important for reducing hallucinations in the reasoning trajectory; thus, we assigned a relatively high weight to λ₂. Locatability, being a binary score that evaluates whether the image and its reasoning are geographically grounded, was assigned a smaller weight (λ₁) to provide auxiliary regularization. We have conducted additional experiments to validate the effect of different weight settings.
>
> ### Table 2.  Performance comparison of weight selection ( λ₁, λ₂, and λ₃) for the GRPO framework on MP16-Reason-Test
> | GRPO | Street (1km) | City (25km) | Region (200km) | Country (750km) | Continent (2500km) |
> | ------- | -------- | -------- | -------- |  -------- | -------- |
> | λ₁, λ₂,  λ₃ = 0.2, 0.5, 1 (proposed setting) | 17.99 | 62.85 | 73.83 | 86.68 | 92.52 |
> | λ₁, λ₂,  λ₃ = 1, 1, 1 | 17.67 | 59.94 | 71.83 | 84.80 | 91.20 |
> | λ₁, λ₂,  λ₃ = 1, 0.5, 0.2 | 17.63 | 59.96 | 72.11 | 84.87 | 91.55 |
>
> The results suggest that the setting λ₁=0.2, λ₂=0.5, λ₃=1, used in our main experiments, is relatively effective and well-balanced for the task.
>
> > ### 4. The potential baselines (W4)
>
> A4: We have additionally included Doubao-1.5-thinking-vision-pro [8] for comparison. While GPT-4.1 and Doubao-1.5-pro [8] still outperform GLOBE in absolute accuracy, they are large-scale proprietary systems with undisclosed architectures and training data. In contrast, GLOBE is fully open-source, data-efficient, and trained on curated supervision, offering a competitive and reproducible alternative for reasoning-based geo-localization. To promote transparency and community advancement, we will release all $\textbf{code}$, $\textbf{models}$, and $\textbf{curated data}$.
>
>
> ### Table 3.  Performance comparison on MP16-reasoning-Test
> | Method | Street (1km) | City (25km) | Region (200km) | Country (750km) | Continent (2500km) |
> | ------- | -------- | -------- | --------- | -------- | -------- |
> | GPT-4.1 | 20.05 | 66.76 | 79.70 | 89.84 | 94.53 |
> | Doubao-1.5-thinking-vision-pro [8] | 18.89 | 64.02 | 76.55 | 88.33 | 93.44 |
> | GLOBE (ours) | 17.99 | 62.85 | 73.83 | 86.68 | 92.52 |
>
>
> Regarding the model size, the parameter count of GLOBE (based on Qwen2.5-VL-7B) is $\textbf{stated in the caption of Table 2}$ in the paper. We acknowledge that this may not be sufficiently prominent, and in the revision, we will explicitly report the model size (7B) in the main text.
>
> > ### 5. The discussion of privacy concerns (W5)
>
> A5:  We thank the reviewer for raising this important point. While our work does not aim to directly address privacy risks or adversarial robustness (such as [6] and [7]), we acknowledge the broader societal implications of geo-localization systems ([5]). The discussion in Section A.3 was intended as a responsible reflection on potential concerns, rather than a claim of technical contributions in this area.
>
> > ### 6. The clarification of annotations among different datasets (Q1)
>
> A6: Different annotations are provided in Table 4, showing that our multi-model distillation produces more diverse and independent reasoning paths compared to existing dataset baselines. We use multi-model voting to reduce hallucinations compared with a single model. During GRPO training, the reasoning paths themselves are not used directly; instead, entities extracted from the paths serve as the reward signal. Reviewer wDfp’s Table 1 also provides the performance impact of single- versus multi-model annotation.
>
> ### Table 4.  Comparison of different annotations
> | Dataset | Annotation Sample |
> | - | - |
> | MP16 [4] | “IMG_ID”: “3f_e4_302010632.jpg”,  “LAT, LON”: “40.001393,-83.019803”|
> | MP16-Pro [9] | “IMG_ID”: “3f_e4_302010632.jpg”,  “LAT, LON”: “40.001393,-83.019803”, “neighbourhood,city,county,state,region,country,country_code,continent” = “,,Grand County,Utah,,United States,us,” |
> | MP16-Reason | “IMG_ID”: “3f_e4_302010632.jpg”, “Reason”: “The image shows … which is located in Columbus.”, “city, country” = “Columbus, United States”, “Entities”: [{"text": "stadium", "type": "ARCH"}, ... , {"text": "seating", "type": "ARCH"}] |
>
> > ### 7. The discussion of failure cases (Q2)
>
> A7: We categorize failure cases into two types: Error Reasoning and Right Reasoning, with representative examples shown in Table 5. Analysis reveals two common patterns: (1) visually similar features (e.g., domes, arches) leading to incorrect landmark attribution, and (2) correct reasoning biased toward locations more frequent in the training set.
>
> ### Table 5.  Comparison of failure cases
> | Type | IMG_ID | Ground Truth  | Reasoning | Prediction |
> | - | - |  - | - | - |
> | Error Reasoning  | eb_80_511397613.jpg |  Etterbeek, Belgium | The grand scale of the architecture ... resembles the Louvre Pyramid or the Arc de Triomphe area. | Paris, France |
> | Right Reasoning  | 98_80_2913796353.jpg  |  Şehitkamil, Turkey | The domed roof with a pointed tip ... suggests Ottoman or Islamic architectural style. | Baku, Azerbaijan |
>
> > ### 8. The discussion of robustness (Q4)
>
> A8: To assess robustness, we conducted zero-shot testing on a 3K subset of the out-of-domain OSV-5M [10] dataset (a geolocalization dataset of streetview images). As shown in Table 6, GLOBE demonstrates strong generalization and outperforms all open-source baselines, including both LVLMs and non-LVLMs.
>
> ### Table 6.  Performance comparison on OSV-5M [10] (mini-3K)
> | Method | Street (1km) | City (25km) | Region (200km) | Country (750km) | Continent (2500km) |
> | - | - | - | - | - | - |
> | ISNs | 0.00 | 1.07 | 6.77 | 22.04 | 44.01 |
> | GeoCLIP | 0.07 | 1.57 | 13.87 | 44.51 | 73.26 |
> | Qwen2.5-VL-7B | 0.00 | 0.87 | 5.14 | 19.81 | 40.55 |
> | InternVL3-8B | 0.00 | 0.73 | 5.27 | 19.81 | 44.01 |
> | GLOBE (ours) | 0.00 | 1.87 | 14.04 | 45.01 | 74.16 |
>
>
> [1] LESS: selecting influential data for targeted instruction tuning. ICML 2024
> [2] LIMA: Less Is More for Alignment, NeurIPS 2023
> [3] Jiuzhang3.0: Efficiently improving mathematical reasoning by training small data synthesis models, NeurIPS 2024
> [4] The benchmarking initiative for multimedia evaluation: Mediaeval 2016. IEEE MultiMedia
> [5] Granular Privacy Control for Geolocation with Vision Language Models, arXiv 2024
> [6] Images are Achilles’ Heel of Alignment: Exploiting Visual Vulnerabilities for Jailbreaking Multimodal Large Language Models, ECCV 2024
> [7] Agent Smith: A Single Image Can Jailbreak One Million Multimodal LLM Agents Exponentially Fast, ICML 2024
> [8] Seed1.5-VL Technical Report, arXiv 2025
> [9] G3: An Effective and Adaptive Framework for Worldwide Geolocalization Using Large Multi-Modality Models, NeurIPS 2024
> [10] OpenStreetView-5M: The Many Roads to Global Visual Geolocation, CVPR2024

---

> ### Comment · Reviewer_WWdG · 2025-08-07
>
> Thanks for the rebuttal with additional analysis. The results have addressed my concerns, and I have updated the score.

---

### Official Review · Reviewer_wDfp · 2025-07-03

**Clarity:** 3
**Significance:** 3
**Originality:** 2
**Rating:** 5
**Confidence:** 3

**Summary:**

The paper discuss the luck of explainability in current Geo-Location methods , due to the black box nature of current solutions and the nature of current datasets
The proposed method allows both reasoning and better explanbility while achieving on par SOTA results with minimal training data.

The autores have created a new dataset MP16-Reason from social media , with main advantage of reasoning trace.
It was done , using  three VLMs QwenVL, InternVL, and GeoCLIP and suggested knowledge distillation data curation pipeline.

The suggested GLOBE pipeline, aims to do learning by reasoning rather than previous so called "pixel memorization" methods.
Three task specific reward functions was learned with annotated supervision and used to model fine-tuning by  GRPO-based RL.

**Questions:**

No Questions
I thanks the authors  for their contribution

**Ethical Concerns:**

["NO or VERY MINOR ethics concerns only"]

**Final Justification:**

The authors provided a clear clarification of my concern. I am keeping my original score and believe this paper will make a valuable contribution to research in this field.

**Limitations:**

yes

**Paper Formatting Concerns:**

.

**Quality:**

3

**Strengths And Weaknesses:**

Strengths
- Proving new reach dataset ,with extra dimensionality of labeling , in form of reasoning trace
- Achieve on par results with SOTA while using much less training data - a good evidence to for the generalization ability of the method
- Innovative Multi-model distillation and data curations

Weakness
- Bias by the three VLMs builtin bias
- The origin raw web based data. The advantage of minimal 33K data for training , is also its weakness: luck of richness , generalization , and variety of semantic visual cues , may generate a bias
- Results outperform open source , yet not showing advantage on the closed ones

---

> ### Author Rebuttal · Authors · 2025-07-30
>
> Thank you for your positive feedback and insightful comments. We are greatly encouraged by your recognition of our $\textbf{curated dataset}$, the $\textbf{strong experimental performance}$, and $\textbf{the innovative ideas}$ presented in our work. Although no specific questions were raised, we would still like to address the weaknesses mentioned in your review to further clarify our motivation.
>
> > ### 1. The explanation of bias in three VLMs (W1)
>
> A1: We thank the reviewer for raising this important point regarding potential bias in our data distillation process. The use of three diverse, high-performing teacher models (Qwen2.5-VL-72B, InternVL3-78B, and GeoCLIP) is a deliberate design choice aimed at mitigating model-specific biases. Instead of relying on a single teacher, which may propagate its own systematic preferences or reasoning styles, our multi-teacher framework leverages the consensus and complementarity of multiple SOTA VLMs to enhance the robustness and diversity of the distilled signals [1]. As shown in Table 1, we compare performance using a single VLM versus our proposed multi-VLM strategy. Across both SFT and GRPO settings, distillation from three VLMs consistently yields better results, indicating improved training data quality and reduced bias compared to using only one VLM.
>
> ### Table 1. Performance comparison on MP16-reasoning-Test
>
> | BackBone | Data Distillation Strategies | Training Strategies | Street (1km) | City (25km) | Region (200km) | Country (750km) | Continent (2500km) |
> | ------- | -------- | -------- | --------- | -------- | -------- | --------- | --------- |
> | Qwen2.5-VL-7B | Baseline | - | 15.42 | 52.72 | 62.86 | 75.11 | 83.47 |
> | Qwen2.5-VL-7B | Only InternVL3-78B distill	 | SFT | 15.22 | 52.47 | 65.09 | 78.79 | 86.15 |
> | Qwen2.5-VL-7B | Only InternVL3-78B distill	| GRPO | 17.37 | 59.45 | 71.88 | 84.74 | 91.24 |
> | Qwen2.5-VL-7B| Using 3 VLMs | SFT | 16.38 | 56.76 | 70.21 | 83.82 | 90.75 |
> | Qwen2.5-VL-7B| Using 3 VLMs (GLOBE) | GRPO | 17.99 | 62.85 | 73.83 | 86.68 | 92.52 |
>
> *Here, we report distance-based (Haversine) accuracy computed after converting predicted city names to GPS coordinates via the Microsoft Azure Maps tools.
>
>
> > ### 2. The explanation of bias in MP16-Reason (W2)
>
> A2: We emphasize that our data curation is not a source of bias, but a controlled knowledge distillation process designed to extract high-quality, informative samples for efficient LVLM training. This approach aligns with recent advances in the literature, where large models are used to construct compact, high-signal training sets for downstream tasks [2, 3, 4].
> Crucially, the goal of distillation is not to replicate the full data distribution, but to identify a representative and semantically rich subset that maximizes learning efficiency. By prioritizing quality over quantity, our method enables state-of-the-art performance with significantly reduced training data and cost.
> $\textbf{Moreover, our objective is not to encourage memorization of specific samples}$, but to activate the model’s reasoning capabilities through well-structured, high-signal examples. These curated instances serve as cognitive anchors that guide the model to internalize spatial and contextual reasoning patterns, thereby promoting generalization, leveraging the model’s internal world knowledge even in the presence of limited visual diversity.
>
> > ### 3. The results compared to the closed ones (W3)
>
> A3: While our method achieves strong performance among open-source LVLMs, we acknowledge that it does not surpass closed-source models such as GPT-4.1. This gap is not unexpected, as industrial-scale systems benefit from vastly larger training datasets, proprietary infrastructure, and orders-of-magnitude more compute resources, typically beyond the scope of academic research.
> Our goal is not to compete directly with such closed systems, but to advance open, reproducible, and data-efficient LVLM training. Within this context, our approach demonstrates significant progress. Notably, GLOBE, built upon the Qwen2.5-VL-7B backbone, outperforms the much larger Qwen2.5-VL-72B (the very model used to generate the distilled data) on the MP16-Reason-Test and IMG2GPS3k benchmark.
> This "student surpassing teacher" result highlights the effectiveness of our distillation and training framework in extracting and refining high-quality knowledge, rather than merely replicating teacher behavior. It underscores the value of curated, reasoning-grounded supervision in boosting model capabilities beyond scale.
> To promote transparency and community advancement, we will release all $\textbf{code}$, $\textbf{models}$, and $\textbf{curated data}$.
>
> [1] MTMS: Multi-teacher Multi-stage Knowledge Distillation for Reasoning-Based Machine Reading Comprehension, SIGIR 2024
> [2] LESS: selecting influential data for targeted instruction tuning. ICML 2024
> [3] LIMA: Less Is More for Alignment, NeurIPS 2023
> [4] Jiuzhang3.0: Efficiently improving mathematical reasoning by training small data synthesis models, NeurIPS 2024

---

> > ### Comment · Reviewer_wDfp · 2025-08-04
> > **Clarify dataset bias vs resource scaling**
> >
> > I appreciate the authors' efforts in addressing the concerns I raised.
> > However, I notice a potential contradiction between your response to Weakness 2 (the use of a small, biased dataset) and your explanation for Weakness 3 (inferior results compared to closed models), which attributes the gap to limited compute resources.
> > Could the authors clarify how they expect their method’s performance to improve with access to greater resources? Specifically, would the gains primarily come from incorporating more data, or are there other contributing factors?

---

> > > ### Author Response · Authors · 2025-08-05
> > >
> > > We sincerely thank the reviewer for your thoughtful feedback. We are grateful for your recognition of our efforts to address the initial concerns.
> > >
> > > Regarding your comment on the relationship between our emphasis on data efficiency (W2) and the performance gap with closed-source models due to resource limitations (W3), we appreciate this insightful observation. We clarify that our method is not a compromise under limited resources, but a scalable paradigm for quality-driven learning.
> > >
> > > Unlike closed models that rely on massive data and compute, our approach prioritizes training efficiency through high-quality, reasoning-aware supervision. The smaller dataset is not a limitation, but a deliberate design: we distill diverse and consistent signals from multiple strong teachers to ensure each example maximizes learning value. This enables the student model to develop robust spatial reasoning with fewer, but more informative, samples.
> > >
> > > To your question of how performance would improve with greater resources: while more high-quality data would naturally help, the gains would not come primarily from quantity, but from scaling the quality pipeline itself: **Larger-scale distillation** (e.g., applying our multi-teacher framework to broader, globally representative image collections) and **Integration with advanced training** (e.g., using the distilled data as foundation for RL with visual reasoning rewards).
> > >
> > > Thus, our method does not oppose scaling, it makes scaling more effective. We shift the focus from how much data is used to how well knowledge is transferred. This offers a reproducible, open, and cognitively grounded path toward efficient LVLM advancement.

---

> > > ### Author Response · Authors · 2025-08-09
> > >
> > > As the rebuttal discussion period will conclude in less than three hours, I just wanted to kindly confirm whether you have had a chance to review our latest responses. If everything looks fine, please feel free to click the “Acknowledgement” button at your convenience.
> > >
> > > Thank you again for your time and effort in reviewing our work.

---

### Note · Authors · 2025-08-12

We thank all reviewers for their time and insightful feedback, and appreciate their recognition of our **novel dataset and methodology** (MP16-Reason with multi-VLM reasoning-grounded distillation), **effective training strategy** (GRPO with locatability, grounding consistency, and geolocation rewards), **strong results with high data efficiency**, **thorough ablations**, and **commitment to open-source release**.

In this rebuttal phase, we addressed the main concerns:
- First, we clarified that our multi-VLM distillation is designed to capture generalizable reasoning patterns rather than reproducing teacher-specific biases, with cross-model experiments confirming its robustness.
- Second, regarding the reward weights ($\lambda_1$, $\lambda_2$, $\lambda_3$), we explained that they were fixed based on task intuition, supported by sensitivity experiments, and demonstrated mathematically that under GRPO normalization, only the relative ratios influence learning, offering guidance for future extensions.
- Third, in comparison with closed-source systems, we emphasized that our objective is to provide an open, efficient, and reproducible alternative, with the “student surpassing teacher” effect illustrating the potential of quality-driven scaling even with limited resources.
- Finally, we report results using average Haversine distance for finer-grained evaluation and will present all main results in this format in the final version, ensuring consistency with the originally reported comparisons.

We are committed to refining the manuscript, including clarifying cross-model distillation results, better framing open- versus closed-source comparisons, standardizing metrics with average Haversine distance, strengthening methodological descriptions, adding inference efficiency comparisons, and updating the introduction and related work sections for clarity and consistency.

We believe these revisions address the key concerns and further strengthen the novelty, robustness, and impact of the work. We sincerely thank the reviewers for their constructive engagement, which has been invaluable in improving our paper.

---

### Decision · Program_Chairs · 2025-09-17

**Decision:**

Accept (poster)

**Comment:**

Summary: Clean approach to produce interpretable geo-reasoning that beats open-source baselines on MP16-Reason-Test, IM2GPS3K, and an OSV-5M mini split (mainly at city+ scales).

Strengths: Novel reasoning-supervised dataset.  Clear GRPO formulation with composite rewards.  interpretability and consistent gains across backbones/datasets.

Weaknesses: Potential circularity/bias from teacher distillation, weak street-level accuracy, initially missing baselines/metrics/details (this was partly fixed during rebuttal).  Based on GPT 4.1 which is a closed model.

(d) Accept (poster): public dataset and GRPO training materially improve open-source LVLM geolocalization with transparent reasoning and solid ablations.  This outweighs residual concerns on bias, fine-grain accuracy, and closed-source gaps.

(e) Rebuttal added Haversine (5-level) metrics, scene breakdowns, cross-model distillation and CoT/λ ablations with GRPO-normalization rationale, extra baselines (PLONK, OSV-5M hybrid).